# MLIP Arena: Advancing Fairness and Transparency in Machine Learning Interatomic Potentials via an Open, Accessible Benchmark Platform

**Yuan Chiang**[1,2,*]    **Tobias Kreiman**[1]    **Christine Zhang**[1]    **Matthew C. Kuner**[1,2]
**Elizabeth Weaver**[1]    **Ishan Amin**[1]    **Hyunsoo Park**[3]    **Yunsung Lim**[4]
**Jihan Kim**[4]    **Daryl Chrzan**[1,2]    **Aron Walsh**[3]    **Samuel M. Blau**[2]
**Mark Asta**[1,2]    **Aditi S. Krishnapriyan**[1,2,*]

[1]UC Berkeley    [2]LBNL    [3]Imperial College London    [4]KAIST

`{cyrusyc,aditik1}@berkeley.edu`

## Abstract

Machine learning interatomic potentials (MLIPs) have revolutionized molecular and materials modeling, but existing benchmarks suffer from data leakage, limited transferability, and an over-reliance on error-based metrics tied to specific density functional theory (DFT) references. We introduce MLIP Arena, a benchmark platform that evaluates force field performance based on physics awareness, chemical reactivity, stability under extreme conditions, and predictive capabilities for thermodynamic properties and physical phenomena. By moving beyond static DFT references and revealing the important failure modes of current foundation MLIPs in real-world settings, MLIP Arena provides a reproducible framework to guide the next-generation MLIP development toward improved predictive accuracy and runtime efficiency while maintaining physical consistency. The Python package and online leaderboard are available at `https://github.com/atomind-ai/mlip-arena`.

## 1 Introduction

The accurate prediction of molecular and material properties has driven innovation for decades and remains crucial for addressing challenges in energy technology, climate change, and drug discovery. While first-principles electronic structure methods have long served as the primary workhorse for property prediction, their computational cost remains prohibitive for scaling atomistic modeling beyond hundreds of atoms. Machine learning interatomic potentials (MLIPs), trained on extensive databases comprising millions of density functional theory (DFT) calculations, have emerged as an efficient and accurate alternative. These models have demonstrated remarkably accurate approximations of the DFT potential energy surface (PES)—the high-dimensional landscape that maps atomic configurations to their corresponding energies and forces—across a wide range of chemical compositions at a fraction of the computational cost of direct DFT evaluations.

Despite excelling in error-based metrics for bulk systems [1], MLIPs trained on the DFT total energy and interatomic forces do not necessarily capture the correct dynamic interactions of atomistic systems [2]. Analogously, classical force fields [3] fit to describe near-equilibrium radial distribution functions cannot capture the energetics of bond-breaking. These limitations may also extend to MLIPs predominantly trained on near- or on-equilibrium configurations. In particular, energy and force regression metrics based on near-equilibrium structures may not reflect performance in downstream scientific tasks. We highlight some specific limitations below.

39th Conference on Neural Information Processing Systems (NeurIPS 2025) Track on Datasets and Benchmarks.

First, energy and force regression metrics are vulnerable to data leakage, failing to accurately assess a model's extrapolation and generalization capabilities. This issue is evident in Matbench Discovery [1], where non-compliant models rank highly for crystal stability metrics due to energy overfitting at the expense of forces and finite-temperature capabilities. This may result in poor generalization to structures more diverse in chemistry and away from the energy convex hull. Additionally, high-ranking models often rely on large datasets, risking test set contamination without proper safeguards.

Second, benchmarks tied to specific datasets or DFT functionals lack flexibility in a rapidly evolving field, where larger, more chemically diverse, or higher-accuracy datasets frequently emerge [4–7]. Static dataset benchmarks quickly become outdated and misleading as newer models trained on larger or proprietary datasets are introduced.

Third, conventional error-based regression metrics often fail to reflect the practical utility and generalizability of MLIPs in real-world applications. Póta et al. [8] recently demonstrated that while some MLIPs exhibit zero-shot capabilities for lattice thermal conductivity prediction, many top-ranked Matbench Discovery models perform worse due to broken crystal symmetry and rough PES derivatives. This underscores that relying solely on regression metrics while ignoring physical priors can widen the gap between model predictions and experimental observables.

To address these challenges, we introduce **MLIP Arena**, a fair and transparent benchmarking platform for foundation MLIPs. This platform evaluates both the quality of the learned PES and the extent to which models respect the physical laws and symmetries essential to atomistic modeling. Unlike previous error-based DFT reference benchmarks [1, 9–12], **MLIP Arena focuses on assessing physical soundness to better evaluate the utility of MLIPs for downstream applications**. By moving beyond error-centric evaluations, it provides more actionable insights for model development and training. Specifically, we assess how well foundation MLIPs capture physics-informed phenomena, their reliability for accurate atomistic simulations, and their readiness for practical scientific research and discovery. The MLIPs evaluated in this work are listed in Table S4.

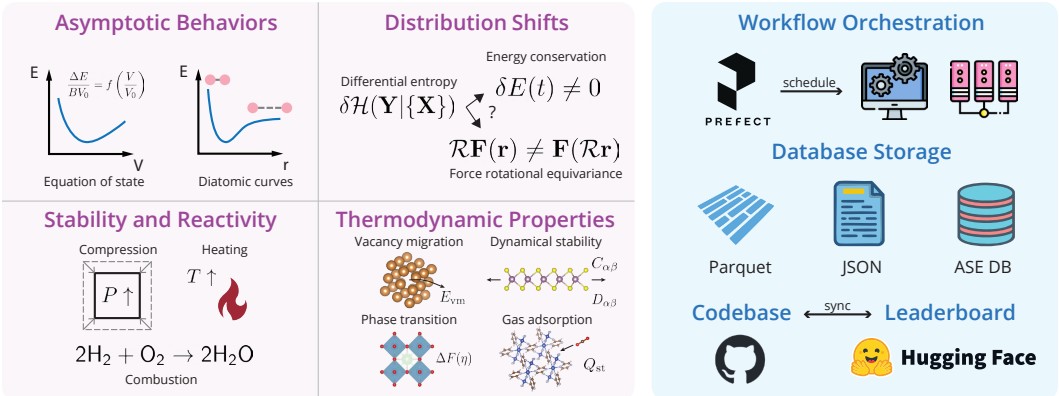

Figure 1: Overview of MLIP Arena. Four benchmark categories beyond error-based regression metrics provide actionable insights agnostic to underlying model architecture and DFT reference. Tasks are defined as Prefect (https://www.prefect.io/) workflows to enable advanced task caching, chaining, and parallel/concurrent execution on HPC. Atomic simulation environment (ASE) [13] calculator and database are used. Codebase (https://github.com/atomind-ai/mlip-arena) and online leaderboard on Hugging Face Space (https://huggingface.co/spaces/atomind/mlip-arena) are available.

## 2 MLIP Arena benchmarks

MLIP Arena assesses the limitations of MLIPs through four primary perspectives. In Section 2.1, we focus on the asymptotic behaviors of MLIP predictions and propose metrics that enable robust and well-balanced ranking from multi-rank aggregation, reducing susceptibility to overfitting on any single metric. Section 2.2 tests MLIP robustness and reactivity under extreme conditions using molecular dynamics (MD) simulations, exposing their instabilities and unphysical behaviors. Section 2.3 investigate the robustness of MLIP to scenarios with quantified distribution shifts to higher

uncertainty. Section 2.4 assesses the predictive capabilities of MLIPs in determining thermodynamic properties and physical phenomena, which requires multiple model passes, higher-order gradients, and more advanced workflows.

## 2.1 Asymptotic analyses on off-equilibrium conditions

> **Asymptotic Behaviors:** The benchmarks evaluate the asymptotic behavior of MLIP predictions on the equation of state (EOS) of stable crystals derived from WBM structures [14] and on the potential energy curves (PECs) of homonuclear diatomics across the periodic table. The quality of prediction is quantified using physical and geometric measures of PECs (including derivative flips, tortuosity, and Spearman's coefficient), assessed in terms of deviations from physically correct values agnostic to DFT references.

Robust MLIPs should predict reasonable asymptotic behaviors of an atomic system under extreme conditions and symmetry transformations. We specifically focus on the metrics agnostic to underlying DFT functional the model has been trained on and propose physical and geometric measures to assess the general performance of MLIPs. A new suite of metrics to reflect the important aspects of MLIPs for atomistic modeling beyond regression errors is proposed as follows.

### 2.1.1 Metrics

**Smoothness.** The major utility of modern MLIPs is the accurate approximation of DFT PES. High-quality PES should be smooth since DFT, as the ground-state electronic structure theory, predicts smooth PES under the assumption of adiabatic approximation on the lowest-energy Born-Oppenheimer surface. Common training objectives on the energy and force of bulk crystals near equilibrium are subject to many-body error cancellation and do not guarantee smoothness. To quantify this effect, we propose *tortuosity* (eq. (S2)), *energy jumps* (eq. (S3)), and *force/gradient flips* to measure the quality of PES.

Tortuosity measures the arc-chord ratio of potential energy curves (PECs, one dimensional slice of PES) projected in the energy dimension. Smooth PECs with a single equilibrium point, like the Lennard-Jones pairwise PEC, have a tortuosity strictly equal to $1$. Energy jump detects the change in the sign of energy gradients and sums up the discontinuity with neighboring points. The number of force/gradient flips count the times force/gradient changes sign along the slice.

**Short-range repulsion.** Atoms at close distances should experience strong repulsion. We use *Spearman's coefficients* to measure the monotonicity of PECs at short interatomic distances or under high compression. Robust MLIPs should have Spearman's coefficients of energy and force close to $-1$ when approaching the repulsive regime. This metric detects the short-range PES *holes*. The absence of these PES holes and the reasonable repulsion are important for the correct samplings of thermodynamic ensembles essential for correct long-time dynamics and physical property calculations.

**Conservative field.** Conservative forces are important for energy conserving molecular simulations, and non-conservative forces are known to degrade the stability of thermostats [15]. We calculate the *conservation deviation* as the MAE between force and the central difference approximation of the derivative of the energy along the PECs (eq. (S1)). We note that energy conservation is a constraint that can be agnostic of the architecture itself, as the standard way it is enforced is by taking gradients of the predicted potential energy in the loss function.

### 2.1.2 Results

The Birch–Murnaghan equation of state (EOS) (eq. (S4)) [16, 17] describes the relationship between the energy and volume of crystalline solids under external pressure and has been computed at scale for materials in the Materials Project [18]. The detailed EOS curves for a set of representative models—each consisting of 21 sampled points evaluated after ionic relaxation at fixed volume (21,000 ionic relaxation trajectories per model)—are visualized in Figure 2. In Table 1, we present the corresponding metrics and their aggregated rankings to assess the quality of the predicted EOS across diverse crystal structures. Both Figure 2 and Table 1 show that most models exhibit the expected

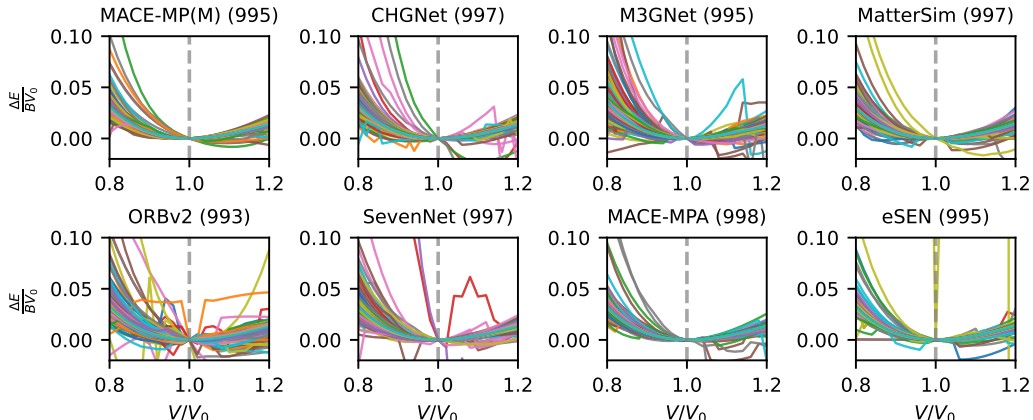

Figure 2: EOS benchmark on 1,000 WBM structures [14]. The reduced relative energy, $\frac{\Delta E}{BV_0}$, is normalized by the bulk modulus $B$ and equilibrium volume $V_0$ through a rearrangement of the Birch–Murnaghan EOS (eq. (S5)). Color indicates the EOS curve of each crystal structure. The number of valid predictions for each model is shown after the model name.

Table 1: Equation of state (EOS) benchmark on 1,000 WBM structures [14]. **Boldface** and underline indicate the **best** and worst metrics across all MLIPs, respectively. Standard deviations are given in parentheses. Derivative flips are ranked by their absolute deviation from 1. For up-to-date models and aggregated rankings, see the online leaderboard.

| Model | Derivative flips ↓ | Tortuosity ↓ | Spearman's coefficient | | | Missing ↓ |
| | | | E: compression ↓ | $\frac{dE}{dV}$: compression ↑ | E: tension ↑ | |
|---|---|---|---|---|---|---|
| MACE-MPA | **1.037 (0.283)** | **1.005 (0.054)** | **-0.999368 (0.012)** | 0.996332 (0.039) | **0.993186 (0.077)** | **2** |
| eSEN | 1.042 (0.314) | 1.008 (0.090) | -0.999330 (0.012) | **0.996857 (0.037)** | 0.992097 (0.073) | 5 |
| MACE-MP(M) | 1.042 (0.345) | 1.009 (0.129) | -0.999330 (0.011) | 0.994116 (0.059) | 0.991586 (0.088) | 5 |
| MatterSim | 1.045 (0.376) | 1.006 (0.055) | -0.997350 (0.039) | 0.992790 (0.078) | 0.988098 (0.115) | 3 |
| CHGNet | 1.105 (0.540) | 1.015 (0.123) | -0.996499 (0.051) | 0.992997 (0.052) | 0.986642 (0.117) | 3 |
| SevenNet | 1.109 (0.555) | 1.019 (0.275) | -0.998128 (0.026) | 0.988912 (0.077) | 0.985958 (0.117) | 3 |
| M3GNet | 1.175 (0.676) | 1.018 (0.149) | -0.996321 (0.052) | 0.989743 (0.065) | 0.980169 (0.133) | 5 |
| ORBv2 | 1.316 (0.870) | 1.037 (0.215) | -0.991846 (0.082) | 0.970143 (0.132) | 0.963746 (0.198) | 7 |

concave-up behavior for the majority of structures, although some models display characteristic failure modes, including short-range holes, shifted energy minima, and spurious spikes.

We further perform energy–volume scans under more extreme volumetric strains, ranging from $-49\%$ to $75\%$, but *without* initial structure or ionic relaxation; that is, the fractional coordinates of ions remain fixed after deformation. See appendix A.4 for further analysis. To evaluate whether the models truly capture the underlying interactions, we also analyze the potential energy curves (PECs) of homonuclear diatomics, with interatomic distances spanning $0.9$ times the covalent radius $r_{cov}$ to $3.1$ times the van der Waals radius $r_{vdw}$ across the entire periodic table. This range approximately covers the equilibrium covalent bond length ($2r_{cov}$) and the decay of dispersion interactions (see appendix A.2 for details). Interestingly, many top-ranked models on bulk crystal EOS and Matbench Discovery perform poorly in pairwise interactions, suggesting that apparent benchmark success may result from plausible many-body error cancellation in bulk systems. See Table S1 for rankings and Figure S1 for select PECs in appendix A.2. We encourage reader to visit our interactive leaderboard for complete set of all elements.

## 2.2 Stability and reactivity from molecular dynamics simulation

**Stability and Reactivity:** Isochoric–isothermal (NVT) molecular dynamics (MD) simulations with a temperature ramp from $300\,\mathrm{K}$ to $3000\,\mathrm{K}$ over $10\,\mathrm{ps}$, and isobaric–isothermal (NPT) MD simulations from $300\,\mathrm{K}$ at $0\,\mathrm{GPa}$ to $3000\,\mathrm{K}$ at $500\,\mathrm{GPa}$ over $10\,\mathrm{ps}$, are performed on

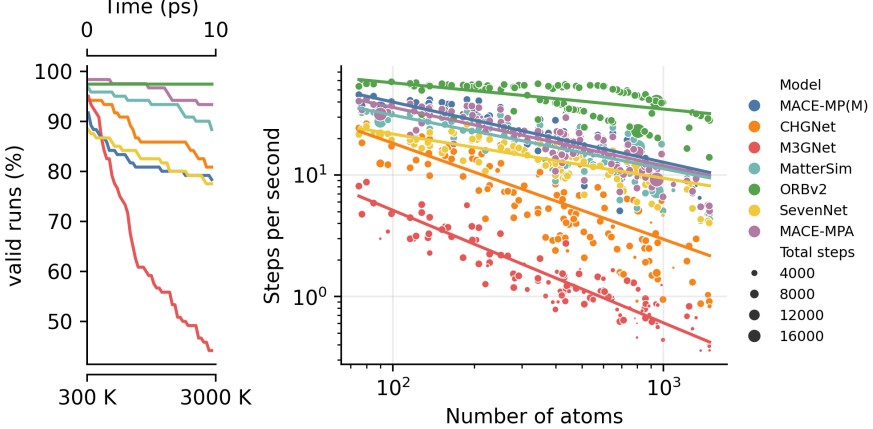

(a) 120 NVT MD simulations from 300 K to 3000 K.

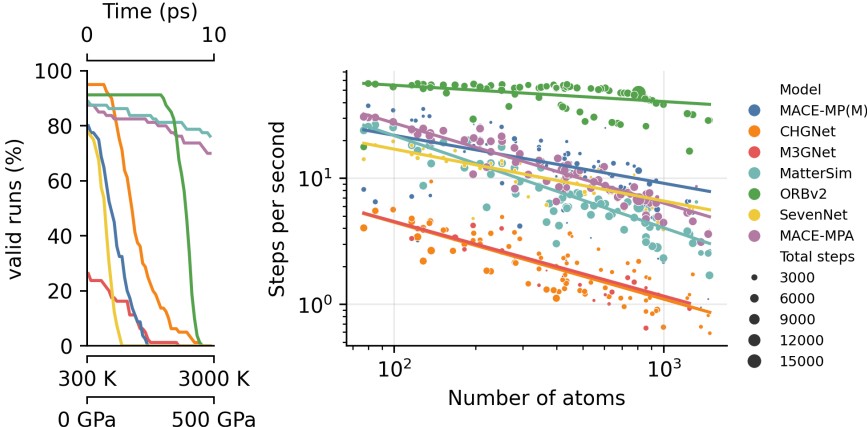

(b) 80 NPT MD simulations from 300 K to 3000 K and 0 GPa to 500 GPa.

Figure 3: MD stability on RM24 structures. For NVT (a), we perform Nosé-Hoover thermostats with linearly increasing temperature from 300 K to 3000 K. The number of valid trajectories and the scaling of MD steps per second (SPS) with the number of atoms $N$ are shown. For NPT (b), Nosé-Hoover thermostats is performed with an additional pressure ramp from 0 GPa to 500 GPa. The size of each point represents the valid steps along each valid trajectory. The power law SPS $= aN^b$ is used to determine the asymptotic performance of MLIPs (solid line). First 120 structures from RM24 are used for NVT, and first 80 structures are used for NPT. The target length of each trajectory is 10 ps. `cuEquivariance` kernel was disabled for MACE family models.

random mixture structures (RM24). Reactivity is exemplified by annealing MD simulations of hydrogen combustion. Benchmarking metrics include the fraction of valid runs and runtime speed performance (MD steps per second on a single A100 GPU).

Stable and accurate MD simulations are essential for atomistic modeling. As their name suggests, MLIPs should serve as reliable interatomic potentials for running MD simulations. We benchmark MLIPs for stability under extreme temperature and/or pressure conditions and record their runtime performance using the random amorphous mixture structure database RM24 (appendix A.5).

**Stability under extreme conditions.** We perform MD simulations on RM24 structures with a linear temperature schedule from 300 K to 3000 K for 10 ps using Nosé-Hoover NVT thermostats [19]. The number of valid runs and asymptotic speed scaling with the system size are presented in Figure 3a. Because many MLIPs exhibit short-range holes or require increasingly large neighbor lists under high

pressure, we additionally apply a linearly increasing pressure from $0\,\mathrm{GPa}$ to $500\,\mathrm{GPa}$ over $10\,\mathrm{ps}$ using Nosé–Hoover NPT barostats (fig. 3b). An MD step is considered valid if the atomic structure exists and has finite energy prediction. This is a relatively lenient criterion when treating MLIPs purely as autoregressive samplers, as it does not account for thermodynamic drifts, fluctuations, or potential structural instabilities. Our benchmark challenges the common belief that equivariant models such as MACE and SevenNet are generally slower than non-equivariant models such as CHGNet and M3GNet. In reality, model architecture, engineering optimizations, and checkpoint quality all contribute to overall MD runtime performance, while the speed and stability of MD trajectories also depend on the chemical system. This is illustrated by the fact that MatterSim shares the same architecture as M3GNet but is significantly more stable and performant. ORBv2 is the fastest and has the best scaling exponent among the tested MLIPs in both heating and compression simulations. However, without an explicit short-range core repulsive potential built in, ORBv2 and many earlier MLIPs could not sustain high-pressure conditions up to $500\,\mathrm{GPa}$.

**Chemical reactivity.**   Classical force fields are periled by the inaccurate description of chemical reactions. While a bouquet of reactive force fields [3] has been parametrized to mitigate this limitation, they have shown limited transferability from one system to another. Although MLIPs hold the promise to bypass the limitation, one should not assume the reactivity to be guaranteed from pretraining. As an example to test the reactivity, we perform annealing MD simulation to emulate hydrogen combustion. Hydrogen combustion is a challenging out-of-distribution (OOD) test since there are multiple bond breaking and formation events that are poorly represented in most of the available MLIP training sets to date [20]. We evaluate the select models on $1\,\mathrm{ns}$ annealing MD simulations ($2 \times 10^6$ steps with $0.5\,\mathrm{fs}$ timestep) by heating a system of hydrogen and oxygen molecules linearly from $300\,\mathrm{K}$ to $3000\,\mathrm{K}$, holding at $3000\,\mathrm{K}$, and then cooling back to $300\,\mathrm{K}$. Temperature fluctuations, number of water molecules, and enthalpy change $\Delta H$ are monitored along MD trajectories (fig. S3). Our results show that the model reactivity is uncorrelated with the prediction accuracy on bulk crystals. See appendix A.7 for detailed comparison between models.

## 2.3   Robustness to distribution shifts

> **Distribution shifts:** The benchmarks assess violations of energy conservation in MLIP MD trajectories and of rotational equivariance in static force predictions under input distribution shifts, characterized by the differential entropy of atomic local environments. Energy drifts are monitored over eight MD trajectory windows to evaluate conservation, while rotation-induced force errors are computed and averaged within bins defined by differential entropy.

While model architectures that strictly adhere to known symmetries and physical laws have been the standard, recent models [21–23] have shown competitive performance with non-conservative and non-equivariant force predictions. While models with fewer constraints adhere to symmetries well in-distribution [21, 23], it is important to understand how these models generalize to out-of-distribution systems when considering them for practical use [24]. To this end, we propose an evaluation to measure robustness to symmetries in the face of out-of-distribution structures.

**Measuring distribution shifts with differential entropy.**   To quantify how far a system is from the training distribution, we compute the differential entropy $\delta\mathcal{H}$ for each structure with respect to the training distribution, as the implemented in the QUESTS descriptors proposed by Schwalbe-Koda et al. [25] (see appendix A.9 for details). The differential entropy provides a measure of uncertainty or "surprise" for one to probe how current MLIPs maintain energy conservation and rotational equivariance in the face of distribution shifts.

**Energy conservation.**   We perform $5\,\mathrm{ps}$ NVE simulations with a $1\,\mathrm{fs}$ time step, initializing atomic velocities from a Maxwell-Boltzmann distribution at 1000 K. Simulations are conducted on random subsets of each model's training set. Differential entropy is computed for structures along the simulation trajectories using a sliding window approach. For each $500\,\mathrm{fs}$ window, the differential entropy of the midpoint structure is calculated with respect to MPTrj, and the energy difference between the start and end of the window is recorded.

Figure 4 shows that direct force prediction models such as ORB and Equiformer demonstrate a significant correlation between higher differential entropy and greater energy deviation, indicating

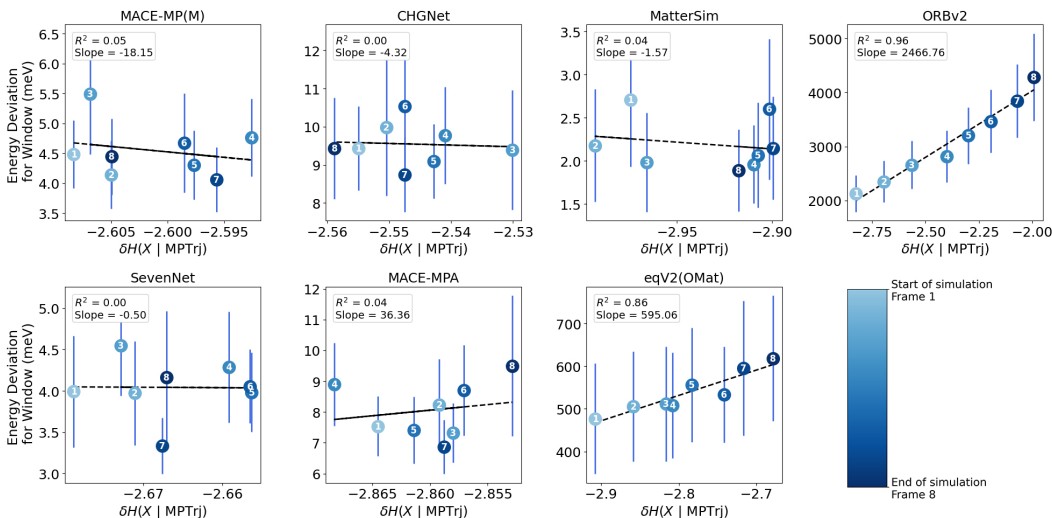

Figure 4: Energy conservation under distribution shift. Energy deviation is calculated for each sliding window during NVE MD simulations for $5\,\text{ps}$. Differential entropy of the structure in the middle of the window is calculated, and the energy deviation from the start to the end of the window is recorded. We report 95% confidence interval error bars and a line of best fit. The order in which windows appear during the simulation is annotated by the number on each point. For direct force prediction models, the simulated trajectories become increasingly surprising over time, as shown by the monotonically increasing numbers from left to right.

that non-conservative models tend to (increasingly) violate energy conservation on structures that are surprising in their training set. We also find that direct force prediction models reach more surprising regions of phase space over the course of simulation, indicated by the increasing window numbers as the differential entropy increases in Figure 4. However, gradient-based force prediction models show little correlation between differential entropy and energy conservation ability. Unlike non-conservative models, gradient-based models do not show increasing surprise as the simulation progresses.

**Force rotational equivariance.** To evaluate the ability of models to learn rotational symmetries from data, we perform a test to quantify learned rotational equivariance. For a rotation matrix $\mathbf{R}$ and atomic positions as $\mathbf{r}$, we measure the MAE between rotated force predictions $\text{MAE}(\mathbf{F}) = \frac{1}{3}\sum_{i=1}^{3}|\mathbf{R}\mathbf{F}(\mathbf{r})_i - \mathbf{F}(\mathbf{R}\mathbf{r})_i|$, where $\mathbf{F}(\mathbf{r})$ represents the models' force predictions for atomic positions $\mathbf{r}$. Perfect equivariance would result in a MAE of 0.0 regardless of the rotation angle.

We evaluate models across a random subset of MPTrj [26], a dataset that consists of inorganic bulk materials. We uniformly sample 500 systems and their trajectories from the dataset. We then calculate the force MAE per frame averaged over 10 random rotation axes and 5 angles from 30° to 180°. Figure S6 shows that the non-rotationally equivariant ORB and ORBv2 models [23] exhibit strong correlation between greater differential entropy and higher rotational force MAE. This indicates that while current non-equivariant architectures can adhere to rotational equivariance on in-distribution structures, they may struggle to maintain symmetries for OOD structures with diverse orientations. The rest of the models, which have rotational equivariance built explicitly into the architecture [27–30], achieve perfect rotational equivariance, as expected.

## 2.4 Thermodynamic properties and phenomenological studies

**Thermodynamic Properties:** This section provides various benchmarks relevant for downstream property applications and phenomenological studies: vacancy formation and migration from nudged elastic band calculations [31], $CO_2$ adsorption for metal-organic frameworks

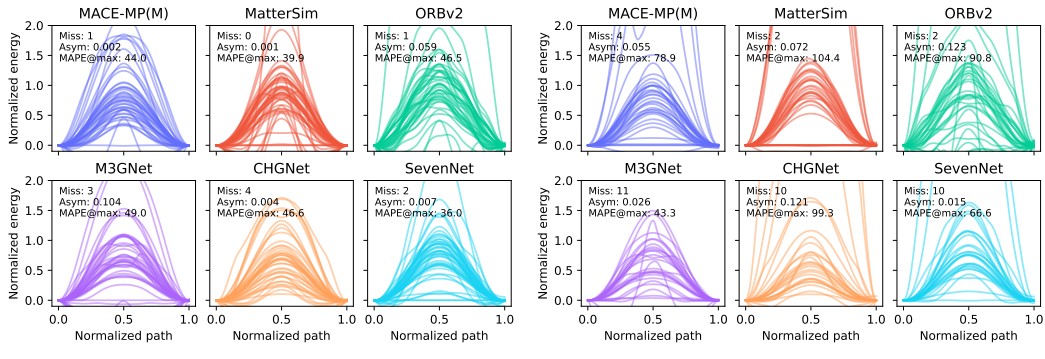

(a) FCC elemental crystals        (b) HCP elemental crystals

Figure 5: NEB profiles of vacancy migration in FCC (a) and HCP (b) elemental crystals. All path lengths are normalized to 1, and all energies are normalized by PBE vacancy migration energy barrier $E_{\mathrm{vm}}^{\mathrm{PBE}}$ as given in [31]. Number of missing predictions, average path asymmetry, and MAPE of maximum energy barrier are annotated on top left.

[32], second-order phase transition in perovskite [33], and dynamical stability screening of 2D materials from C2DB database [34].

**Vacancy formation and migration energies.** Defects, especially vacancies, play a key role in determining the properties of many functional materials used for photovoltaic, catalytic, thermoelectric, and optoelectronic applications [35, 36]. We evaluated six widely used MLIPs capable of predicting stress in elemental face-centered cubic (FCC) and hexagonal close-packed (HCP) crystals, leveraging the vacancy diffusion database by Angsten et al. [31]. The translational symmetry of crystal sites and vacancies requires that the paths and barriers for forward and backward vacancy migration be identical, making this a robust test of a model's ability to respect crystal symmetry.

Climbing image nudged elastic band (CI-NEB) calculations were performed to analyze vacancy migration barriers. We define *path asymmetry* (eq. (S8)) and *barrier asymmetry* (eq. (S9)) of the migration energy profiles in Appendix A.8. We found the symmetry of NEB profiles has no strong correlation with built-in equivariance or not, and in general all models perform worse for HCP crystals. Figure 5 presents the NEB energy profiles of vacancy migration in FCC and HCP elemental solids. HCP pathways are chosen to be on basal plane to avoid asymmetrical migrations. We found that MACE-MP(M), MatterSim, and ORBv2 generally relax NEB more robustly than M3GNet, CHGNet, and SevenNet. MatterSim, MACE-MP(M), CHGNet, and SevenNet exhibit near-perfect mirror symmetry around the saddle point for most FCC paths, while MACE-MP(M) achieves the best balance between symmetry and robustness for HCP paths.

**Extended case studies.** In Appendix A.10, we further assess the downstream utility of MLIPs for three extended case studies: $CO_2$ adsorption in metal-organic frameworks (MOFs) (appendix A.10.1), dynamical stability of 2D materials (appendix A.10.2), and second-order phase transition in perovskite (appendix A.10.3). Each study exposes the certain weakness of modern MLIPs. We found possibly due to poorly described non-bonded interaction between $CO_2$ molecule and MOFs, many MLIPs deviate at a large degree from experimental adsorption energies and may not be informative enough for MOF virtual screening (appendix A.10.1). Our results on the Landau-like second-order phase transition in $BaZrO_3$ (appendix A.10.3) uncover subtle failure modes (energy degeneracy and asymmetrical PES) of octahedral tilts predicted by M3GNet and ORBv2. These limitations could cause the models unable to reproduce the correct transition behaviors important for exotic functional properties such as superconductivity [33, 37]. Furthermore, despite the recent saturation in the prediction of thermodynamic stability and lattice thermal conductivity of 3D bulk crystals, our results on the dynamical stability of 2D materials (appendix A.10.2) indicate that the discovery rate of stable 2D materials remains poor (highest macro F1 score of 0.420 and 0.412 by MACE family models) and there is still a large performance gap in 2D materials space that the success in 3D materials may not ostensibly translate over.

# 3   Related Work

**Static DFT reference benchmarks.**   Benchmarking of MLIPs has largely centered around static DFT datasets such as QM9 [38], ANI-1 [39], MD17 [40], SPICE [6, 41], MPTrj [26], GMTKN55 [42], and more. While these have enabled rapid progress, they are tied to specific level of electronic structure theory and the data are generally incompatible with one another. Over time, the community has expanded benchmark domains, but the dependency on static DFT references remains [1, 9–12, 43, 44]. The Matbench test suite introduced by Dunn et al. [43] compiles 13 tasks (e.g., formation energies, band gaps, elastic moduli) drawn largely from DFT-computed data. Matbench Discovery [1] leveraged the WBM database [14] as an extension beyond the MP [45] for crystal stability classification. Some other benchmarks rely on specific DFT reference while comparing models trained on incompatible dataset [12, 44, 46, 47]. While targeting to higher level of theory is desirable, the higher-level of theory however may not be equally transferable (*e.g.* coupled-cluster theory describes metallic solids poorly [48]), leading to misleading, non-cross-comparable assessments.

Many models now saturate these test metrics, yet fail to extrapolate to unseen chemistry, strained configurations, or finite-temperature behavior. MLIP Arena complements these efforts by introducing physically grounded tasks that probe model robustness beyond interpolation to a static reference.

**Risk of regression error metrics.**   Standard metrics like MAE and RMSE are known to poorly reflect real-world MLIP utility. Bigi et al. [15], Fu et al. [49] show that models with low force errors may fail to conserve energy in MD, while Póta et al. [8], Loew et al. [50] demonstrate that good regression error metrics on energy and forces does not ensure accurate phonons or thermal transport. Direct force models often violate energy conservation due to the lack of a consistent potential energy surface [15, 51]. These inconsistencies could be due to a *misalignment* between energy and force pre-training objectives and physical properties of interest [52]. Moreover, average errors can obscure large failures in rare but critical configurations. MLIP Arena addresses these gaps by incorporating task-specific, granular evaluations to provide a more faithful measure of model reliability.

# 4   Discussion and Conclusion

**Limitations.**   Traditionally, MLIP training and benchmarks rely on DFT references. This is a computationally cheap way to evaluate models since it only requires a few single-point predictions from the ML model, as opposed to autoregressive benchmarks (*i.e.* MD simulations). We acknowledge that moving away from DFT references makes it harder to directly compare models, at least on in-distribution data present in standard test datasets. However, the central promise of foundation MLIPs is to generalize to OOD systems and phases—in which case accuracy with respect to in-distribution DFT data becomes auxiliary rather than primary.

Analogously, many other areas of machine learning have moved beyond standard regression metrics [53]. Large language models, for example, are increasingly evaluated on practical, task-oriented performance rather than raw perplexity on training-like data. In the same spirit, MLIP Arena is our first attempt to prioritize qualities that are essential to atomistic modeling beyond any DFT reference: symmetry, conservation laws, reasonable asymptotic behaviors and thermodynamic properties that any interatomic potential should satisfy for practical utility.

**Opportunities.**   Reference-agnostic benchmarks like MLIP Arena could motivate and guide new directions in model development and training that explicitly tackle generalization and downstream utility, particularly through reinforcement learning [54], implicit differentiation [55], and test-time training [24] approaches. MLIP Arena provides reproducible workflows that can be scaled for high-throughput reward data generation across a broad range of practically relevant OOD tasks, facilitating the exploration of these training paradigms.

In summary, we present MLIP Arena, an open benchmarking platform that avoids simplistic regression metrics susceptible to error cancellation and instead focuses on evaluating physical awareness and practical utility. Our analysis uncovers some new insights: gradient-based force predictions may exhibit non-conservative behavior; alignment between training dataset size and better model performance is not always guaranteed but depends on design choice; and current MLIPs have not saturated in reactivity and robustness under distribution shifts. MLIP Arena serves as a transparent

and reproducible workflow orchestrator, guiding the development of MLIPs with improved adherence to physical principles, runtime performance, and predictive capability.

# 5 Author Contribution Statement

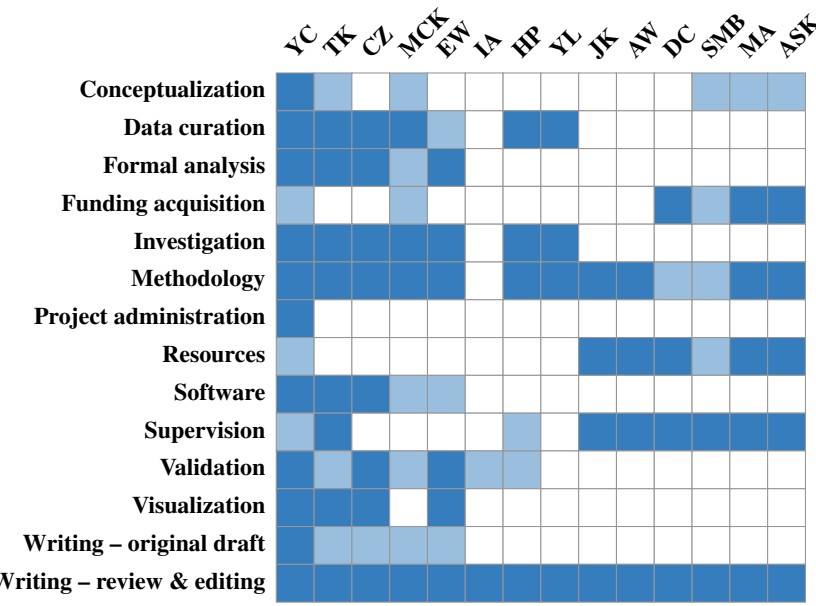

# 6 Acknowledgments

We acknowledge funding through the DOE, Office of Science, Office of Basic Energy Sciences, Materials Sciences and Engineering Division, under Contract No. DE-AC02-05-CH11231 within the Materials Project program (KC23MP). The benchmarks were developed and performed using resources of the National Energy Research Scientific Computing Center (NERSC), a Department of Energy Office of Science User Facility using NERSC award BES-ERCAP0032604. YC received the support from Taiwan-UC Berkeley Fellowship jointly offered by Ministry of Education in Taiwan and UC Berkeley. TK was supported by the Toyota Research Institute as part of the Synthesis Advanced Research Challenge. MCK was supported by the National Science Foundation Graduate Research Fellowship Program under Grant No. DGE-2146752. SMB was supported by the Energy Storage Research Alliance "ESRA" (DE-AC02-06CH11357), an Energy Innovation Hub funded by the U.S. Department of Energy, Office of Science, Basic Energy Sciences.

We thank Aaron Kaplan for advice and comments on the manuscript, and Janosh Riebesell, Philipp Benner, Patrick Huck, Rouxi Yang, Evan Walter Clark Spotte-Smith, and Bowen Deng for early discussions; and Jan Janssen, Rhys Goodall, Abhijeet Gangan, and Han Yang for valuable exchanges.

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

# A   Supplementary Information

## A.1   Details on metrics

**Conservative field.**   Conservative forces are important for physical and stable MD simulations, as extra energy will be injected into or extracted from the system through non-conservative forces, degrading the stability of thermostats [15]. Some models use direct force prediction [22] or apply *post-hoc* correction [23] to achieve better prediction errors or smaller drifts during MD simulations. Despite enhanced speed performance, these non-conservative forces may violate the law of energy conservation, undermining the stability of phonon and MD simulations and the predictive power on finite-temperature thermodynamics quantities [8]. To quantify the deviation of force prediction from the conservative field, we compute the MAE between force and the central difference of energy along the homonuclear diatomic curves:

$$\text{Conservation deviation} = \left\langle \left| \mathbf{F}(\mathbf{r}) \cdot \frac{\mathbf{r}}{\|\mathbf{r}\|} + \nabla_r E \right| \right\rangle_{r=\|\mathbf{r}\|}. \tag{S1}$$

The forces are projected onto the direction of interatomic vectors. Note that this definition is only valid for diatomic interaction but a well-defined, manageable alternative for the exploding combinatorics of hetero-nuclfgear, many-body interactions. Many modern MLIPs have many-body forces, and more careful decomposition of many-body contributions needs to be considered for those cases.

**Short-range stiffness.**   Atoms at close distance should experience strong repulsion. Despite the inaccuracies of DFT calculations at short interatomic distances (appendix A.3), the well-behaved classical force fields and MLIPs should reproduce strong repulsive interactions between atoms at short range distances. In fact, Deng et al. [56] has indicated prominent softening across MLIPs trained on MPTrj, which consists of crystal relaxation trajectories close to equilibrium. Softened potentials often have early drop in energy and forces at the short range, leading to increased probability of instability. To quantify this behavior, we use *Spearman's coefficients* to evaluate the repulsiveness of energy curves (E: repulsion in Table S1) at the distance range $r \in [r_{\min}, r_{\text{eq}}]$, where $r_{\text{eq}} = \underset{r \in [r_{\min}, r_{\max}]}{\arg\min} E(r)$ is taken as the equilibrium internuclear distance. Force curves (F: descending) are evaluated at the distance range between $r_{\min}$ and $\underset{r \in [r_{\min}, r_{\max}]}{\arg\min} F(r)$ where the largest attractive (the most negative) force happens.

**Smoothness.**   The smoothness of a PEC can be heuristically estimated by *tortuosity* as the ratio between total variation in energy $\text{TV}_{r_{\min}}^{r_{\max}}(E)$ and the sum of absolute energy differences between shortest separation distance $r_{\min}$, equilibrium distance $r_{\text{eq}}$, and longest separation distance $r_{\max}$ . This is essentially the arc-chord ratio projected in the energy dimension:

$$\text{Tortuosity} = \frac{\displaystyle\sum_{r_i \in [r_{\min}, r_{\max}]} |E(r_i) - E(r_{i+1})|}{|E(r_{\min}) - E(r_{\text{eq}})| + |E(r_{\text{eq}}) - E(r_{\max})|} \tag{S2}$$

. The Lennard-Jones potential and any potentials with single repulsion-attraction transition or pure repulsion have tortuosity equal to 1. Note that the true PECs of some elements may have intermediate range energy barriers and thus ideally the elemental average across the periodic table should be slightly above one. For the simplicity of this metric, we rank the models by the absolute difference with 1.

We also identify the sign changes of energy gradients on PECs to extract the *energy jump* on both sides to the neighboring sampled points, which can be written down verbatim:

$$\text{Energy jump} = \sum_{r_i \in [r_{\min}, r_{\max}]} \left| \text{sign}\left[E(r_{i+1}) - E(r_i)\right] - \text{sign}\left[E(r_i) - E(r_{i-1})\right]\right| \times$$

$$\left(|E(r_{i+1}) - E(r_i)| + |E(r_i) - E(r_{i-1})|\right) \quad \text{(S3)}$$

. The smoother PEC has lower tortuosity and total energy jump.

## A.2 Homonuclear diatomics

Pairwise interactions are the most important interactions in atomistic systems. PECs have the benefit of being less vulnerable to data leakage as DFT references for PECs are difficult to calculate due to multiple possible spin configurations, basis set incompleteness in local-orbital DFT codes, and convergence issues in plane-wave DFT codes. In Table S1, we compute six physical and geometric measures to rank the homonuclear PECs of MLIPs in three aspects: conservative field, short-range stiffness, and smoothness, as we discuss in the pervious subsection (appendix A.1).

Two atoms are placed inside a vacuum box and the predictions are made with separation distances ranging from 0.9 covalent radius $r_{\text{cov}}$ to 3.1 van der Waals radius $r_{\text{vdw}}$ or to 6 Å if van der Waals radius is not available. The range of interatomic distances is chosen heuristically by the fact that the equilibrium bond length is about the sum of covalent radii (for homonuclear diatomics this is $2r_{\text{cov}}$) and the interatomic energy and forces plateau around $2r_{\text{vdw}}$. We increase the distance range by the factor of 50% of $2r_{\text{cov}}$ and $2r_{\text{vdw}}$ and further extend both ends by 10% of radii. The shortest, equilibrium, and longest separation distances are denoted as $r_{\min}$, $r_{\text{eq}}$, and $r_{\max}$ respectively. **Both energy and force curves are performed** at 0.01 Å interval for dense samplings.

Table S1: PEC quality of homonuclear diatomics based on physical and geometric measures. **Boldface** and underline represent the **best** and the worst metrics across all MLIPs, respectively. Select PECs are shown in Figure S1. Detailed definitions and implementation details are available in Appendix A.1.

| Model | Conservation deviation [eV/Å] | Spearman's coefficient E: repulsion | F: descending | Energy jump [eV] | Force flips | Tortuosity |
|---|---|---|---|---|---|---|
| MACE-MPA | 0.077 | **-0.997** | -0.975 | 0.010 | 1.371 | **1.006** |
| MACE-MP(M) | 0.070 | **-0.997** | -0.980 | 0.038 | 1.449 | 1.161 |
| MatterSim | **0.013** | -0.980 | -0.972 | **0.008** | 2.766 | 1.021 |
| M3GNet | 0.026 | -0.991 | -0.947 | 0.029 | 3.528 | 1.016 |
| ORBv2 | 9.751 | -0.883 | **-0.988** | 0.991 | **0.991** | 1.287 |
| eSCN(OC20) | 2.045 | -0.939 | -0.984 | 0.806 | 0.640 | 5.335 |
| CHGNet | 1.066 | -0.992 | -0.925 | 0.291 | 2.255 | 2.279 |
| ORB | 10.220 | -0.881 | -0.954 | 1.019 | 1.026 | 1.798 |
| SevenNet | 34.005 | -0.986 | -0.928 | 0.392 | 2.112 | 1.292 |
| eqV2(OMat) | 15.477 | -0.880 | -0.976 | 4.118 | 3.126 | 2.515 |
| eSEN | 1.170 | -0.692 | -0.919 | 5.562 | 4.000 | 1.838 |
| ALIGNN | 5.164 | -0.913 | -0.310 | 9.876 | 30.669 | 1.818 |
| EquiformerV2(OC20) | 21.385 | -0.680 | -0.891 | 38.282 | 22.775 | 8.669 |
| EquiformerV2(OC22) | 27.687 | -0.415 | -0.855 | 64.837 | 21.674 | 15.880 |

## A.3 Inaccuracies of PAW DFT calculations at short interatomic distances

Due to the classical treatment of nuclei, frozen core approximation, and smoothed core electron wavefunctions in the projected-augmented wave (PAW) DFT formalism, when two atoms are too close to each other, electron wavefunctions start to overlap and oscillate significantly. In such cases, PAW projectors and plane-wave basis set may not accurately describe core electrons and their interactions with valence electrons, leading to large inaccuracies.

## A.4 Equation of state and energy-volume scan

**Structure selection.** Structures were selected from the WBM dataset [14] with a slight bias to adjust for the elemental imbalance in the original paper. That is, each structure was assigned a

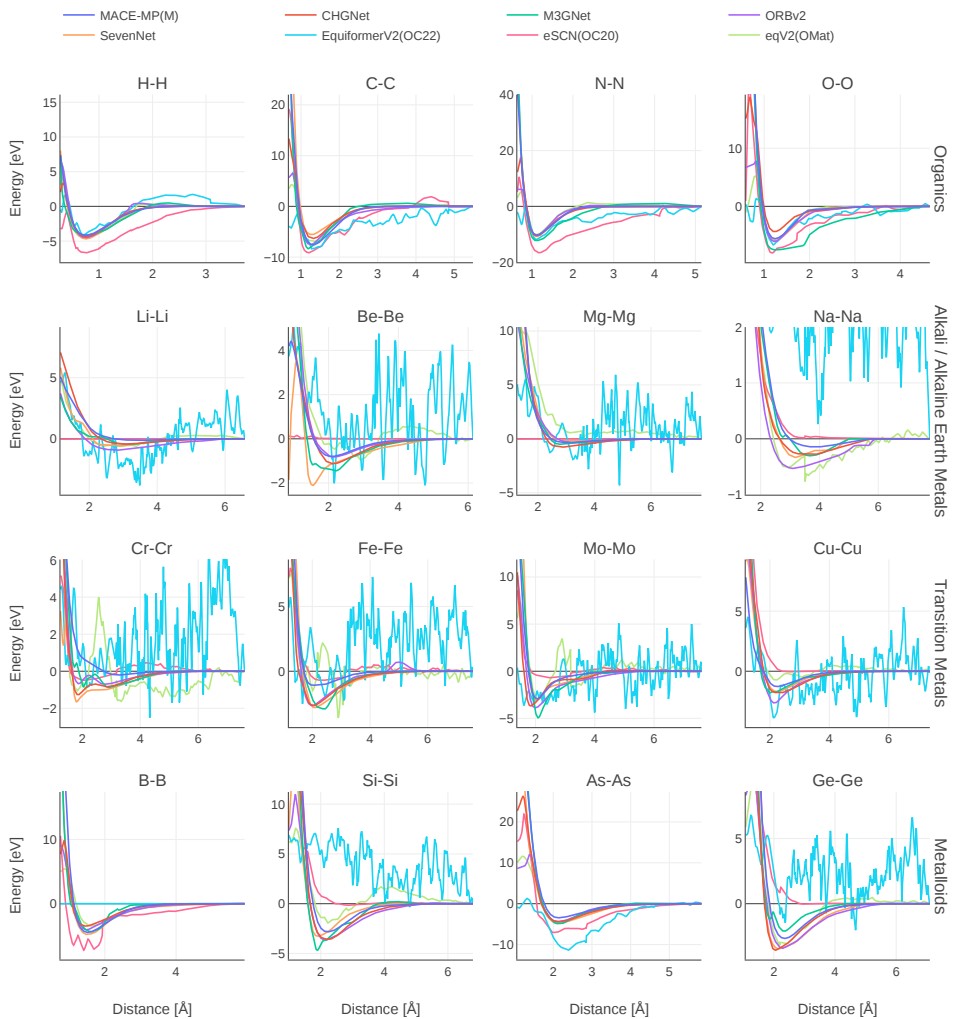

Figure S1: Potential energy curves (PECs) of selected homonuclear diatomic molecules, representing four different chemical characteristics—organics, alkali/alkaline earth metals, transition metals, and metalloids—are presented. The curves from different methods are shifted and aligned to zero at the largest separation distance.

probability for selection based on the prevalence of the elements it contains relative to the overall distribution of elements in the dataset. Elements with lower prevalence in the original dataset were assigned a higher probability of selection; then, 1000 structures from WBM were selected according to these assigned probabilities.

**Equation of state.** The EOS benchmark protocol in Arena includes first unconstrained structure optimization at $0\,\mathrm{K}$ and subsequently multiple energy calculations of isotropic deformations, including ionic relaxation at volumetric strain ranging from $-20\,\%$ to $20\,\%$ of the optimized structure. After ionic relaxation of 21 deformed structures for each crystal, Birch–Murnaghan EOS is fitted with the following equation:

$$E = E_0 + \frac{9BV_0}{16}\left[\left(\eta^2 - 1\right)^2\left(6 + B'\left(\eta^2 - 1\right) - 4\eta^2\right)\right], \quad \eta = \left(\frac{V}{V_0}\right)^{\frac{1}{3}}, \tag{S4}$$

where $V_0$ is the equilibrium volume after initial structure optimization, and $B$ and $B'$ are the bulk modulus and its pressure derivative from the EOS fit. We calculate the reduced relative energy in

Figure 2 by rearranging Equation (S4) as:

$$\frac{\Delta E}{BV_0} = \frac{E - E_0}{BV_0} = \frac{9}{16}\left[\left(\eta^2 - 1\right)^2\left(6 + B'\left(\eta^2 - 1\right) - 4\eta^2\right)\right] \quad (S5)$$

**Energy-volume scan.**   MLIPs should provide reasonable predictions at extreme deformations. In this benchmark, we take 1,000 structures selected from the WBM dataset [14] and uniformly deform them by $\pm 20\%$ along each *lattice* vector (i.e. from 0.51 to 1.73 of the initial volume). The energy of each deformed structure is evaluated *without* relaxation—preventing relaxation to another crystal system. Table S2 presents five metrics to evalaute the performance of MLIPs on the energy-volume scan benchmark. Unlike EOS benchmark in Section 2.1.2, all of the select MLIPs have no missing predictions. Our result shows the saturation of all five metrics for top-ranked models.

In comparison with EOS benchmark, energy-volume scan evaluates the orthogonal performance of MLIPs. Birch–Murnaghan EOS is theoretically defined properties based on finite elastic theory under isothermal condition. EOS benchmark evaluates both model's capability to search for energy minima under relaxation protocol, thus mixing the consequences of energy minimum location and relaxation trajectory. Energy-volume scan tests more extreme condition and especially exposes short-range PES holes, penalizing the models with known corrugated short-range PES where DFT however may not converge as well. Energy-volume scan is also based on the assumption that the WBM structures are at local energy minima, but further investigation reveals that this assumption does not hold universally. There are a few structures with shifted energy local minima consistent across different MLIPs.

**Expected behavior under compression.**   When subjected to significant compression, crystalline materials are expected to exhibit strong short-range repulsion. We evaluate this behavior using the Spearman's rank correlation coefficient to quantify the monotonic increase in energy with decreasing volume. Additionally, the energy derivative $\frac{dE}{dV}$ is expected to steepen in the high-compression regime. MLIPs that are physically consistent should yield Spearman's coefficients approaching $-1$ in the compressive region of the energy–volume curve.

**Expected behavior under tension.**   Under tensile strain, the system's energy should also increase monotonically as atomic bonds are progressively stretched. However, as the crystal approaches dissociation into isolated atoms, the slope of the energy–volume curve ($\frac{dE}{dV}$) is expected to flatten. Thus, we evaluate only the monotonicity of the energy increase under tension using Spearman's coefficients. Reliable MLIPs should produce coefficients close to $+1$ in the tensile regime.

Table S2: Energy-volume scan of 1,000 WBM structures [14]. **Boldface** and underline represent the **best** and the worst metrics across all MLIPs, respectively.

| Model | Derivative flips ↓ | Tortuosity ↓ | Spearman's coefficient | | |
| | | | E: compression ↓ | $\frac{dE}{dV}$: compression ↑ | E: tension ↑ |
|---|---|---|---|---|---|
| eSEN | **1.000000** | **1.000403** | **-0.998339** | **1.000000** | 0.999045 |
| MACE-MPA | **1.000000** | 1.000676 | **-0.998339** | 0.999309 | 0.998718 |
| CHGNet | **1.000000** | 1.000629 | -0.998279 | 0.943964 | **0.999091** |
| MatterSim | 1.009000 | 1.000567 | -0.998097 | 0.999709 | 0.993754 |
| eqV2(OMat) | 1.035000 | 1.000835 | -0.998206 | 0.997224 | 0.998645 |
| M3GNet | 1.002000 | 1.002001 | -0.997588 | 0.997442 | 0.996468 |
| ORBv2 | 1.058000 | 1.004065 | -0.997770 | 0.970752 | 0.997600 |
| SevenNet | 1.034000 | 1.010025 | -0.995164 | 0.946558 | 0.994705 |
| MACE-MP(M) | 1.121000 | 1.080713 | -0.943806 | 0.901188 | 0.998745 |
| ALIGNN | 3.909000 | 1.375652 | -0.889207 | 0.760271 | 0.862085 |

## A.5   Random mixture dataset (RM24)

New materials are often found by reacting two stable materials into a single phase. In the conceptually similar procedure, we generate 1,000 random mixture structures at arbitrary ratio of two stable materials from Materials Project (v2024.12.18). As the binary and ternary compounds are already well covered by MP, we aim at higher component systems of up to six elements from the mixture of binary and ternary systems. All stable binary and ternary materials are first retrieved, totaling 24,430

structures. The number of possible 2-combination is >298M. We randomly selected 1,000 pairs from possible combinations and generate the initial structures using Packmol [57] and Muse [58] to consider periodic boundary conditions. Each generated structure is then relaxed via FIRE optimizer and NVT MD simulation at $1500\,\mathrm{K}$ for $10\,\mathrm{ps}$ with Ziegler-Biersack-Littmark (ZBL) screened nuclear repulsion potential [59]. The final element count distribution of 1,000 structures is presented in Figure S2. The ASE DB file is available at `https://huggingface.co/datasets/atomind/mlip-arena`.

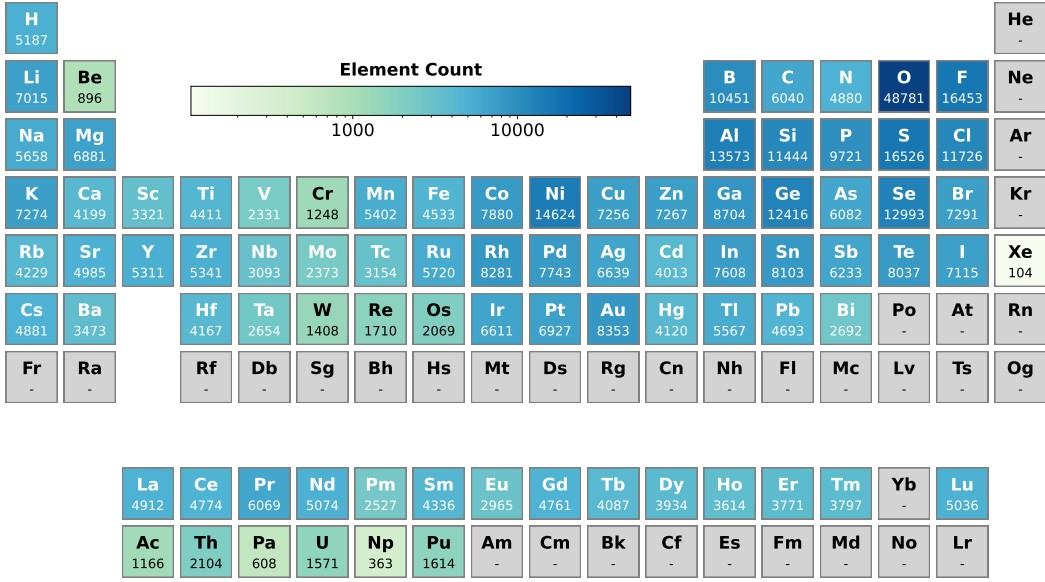

Figure S2: Element counts of random mixture dataset (RM24).

## A.6 MD stability

We performed Nosé-Hoover thermostat and barostat on RM24 structures with linear scheduling of temperature from $300\,\mathrm{K}$ to $3000\,\mathrm{K}$ and/or pressure from $0\,\mathrm{GPa}$ to $500\,\mathrm{GPa}$ across 10 ps MD. The number of valid runs and asymptotic speed scaling with the system size are presented in Figure 3a. Using Prefect (https://github.com/PrefectHQ/prefect) utility, we ensure each run has access to the same resource of 1 AMD EPYC 7763 (Milan) CPU core and 1 NVIDIA A100 (Ampere) GPU. Each run has two retries, with timeout of 600 elapsed seconds for each retry. In section 2.2, the frames are marked as invalid if the simulation cannot reach the timestep or have non-numerical energy values.

## A.7 Hydrogen combustion

CHGNet, EquiformerV2(OC20), eSCN(OC20), and M3GNet were not able to finish $1\,\mathrm{ns}$ MD trajectories (see Figure S3). As analyzed in Figure S4, the slow runtime performance of models without built-in equivariance, such as CHGNet and M3GNet, may seem surprising since equivariant models are often more expensive to use. However, we found that molecules condense into droplets at an early stage in CHGNet and M3GNet trajectories, drastically increasing the number of bond and angle edges and therefore slowing down the MD speed.

While ORB and ORBv2 were fastest in terms of MD steps per second (fig. S4), they could not react hydrogen and oxygen at the elevated temperature and keep the number of water molecules close to zero throughout the trajectories; they also have positive reaction enthalpies, contradicting experimental measurements [60]. Figure S4 also shows that direct force prediction models (EquiformerV2(OC20), ORB) have large center-of-mass drifts ($> 10^2\,\text{Å}$) during MD simulations by six orders of magnitude more than gradient-based models. Enforcing net zero forces as implemented by ORBv2 only decreases the drift to ($\sim 2.4\,\text{Å}$), while other models keep drifts around $10^{-4}\,\text{Å}$ scales over $1\,\mathrm{ns}$ MD.

Here one should note that enforcing net zero forces does not guarantee zero center-of-mass (COM) drift during MD simulations under thermostats. For canonical ensemble like Nosé Hoover thermostats

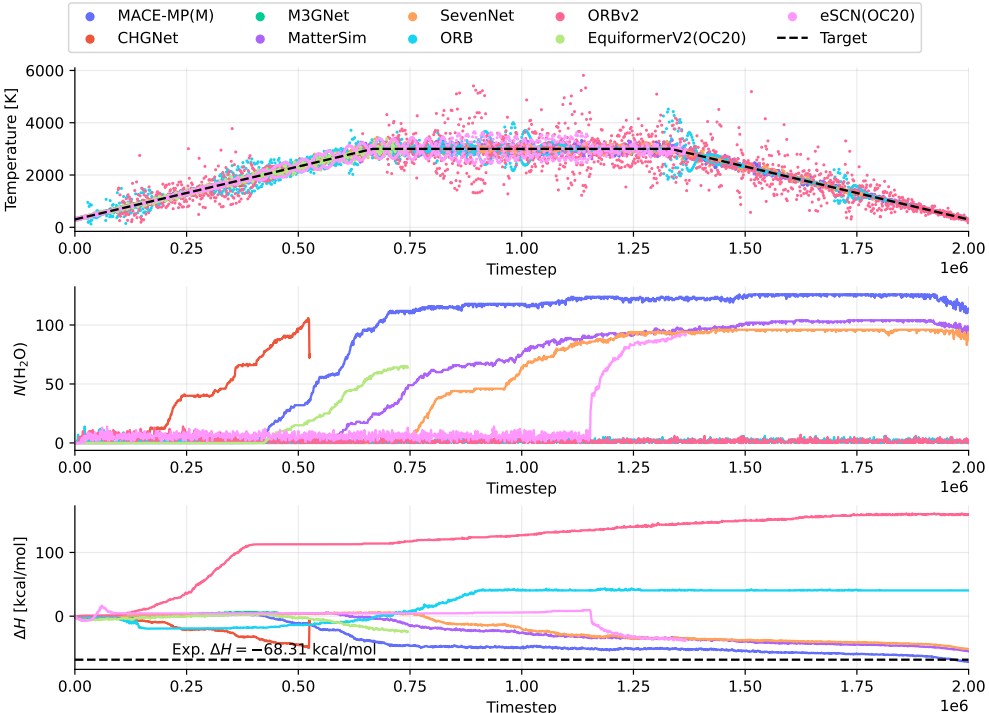

Figure S3: Hydrogen combustion via annealing NVT MD simulation ($128\,H_2 + 64\,O_2 \longrightarrow 128\,H_2O$). Applied temperature schedule is illustrated in the top panel. The experimental reaction enthalpy of $-68.31\,\mathrm{kcal/mol}$ is annotated in the bottom panel [60]. CHGNet, EquiformerV2(OC20), eSCN(OC2), and M3GNet could not finish $1\,\mathrm{ns}$ MD trajectories. Experimental adiabatic flame temperature of hydrogen ranges from $2380\,\mathrm{K}$ (air) to $3000\,\mathrm{K}$ (pure $O_2$) [61]. Only MACE-MP(M) and EquiformerV2(OC20) ignite within this region. Runtime performance and center-of-mass drift are available in Figure S4.

used here [62], the heat bath acts an extra correction on the equations of motion for the system of particles with coordinates $\mathbf{q}_i$, momenta $\mathbf{p}_i$, masses $m_i$, and interaction potential $V$

$$\dot{\mathbf{q}}_i = \frac{\mathbf{p}_i}{m_i}, \tag{S6}$$

$$\dot{\mathbf{p}}_i = -\frac{\partial V}{\partial \mathbf{q}_i} - \mathbf{p}_i \frac{p_\xi}{Q}, \tag{S7}$$

where $\mathbf{p}_\xi$ and $Q$ are the artificial momentum and mass of the thermostat particle [13, 63]. *Post-hoc* correction to enforce net zero force from the model prediction will only correct the first term. The total momentum drift is not zero, as can be seen by the following simple proof:

$$\sum_i \dot{\mathbf{p}}_i = \sum_i \left[ -\frac{\partial V}{\partial \mathbf{q}_i} - \mathbf{p}_i \frac{p_\xi}{Q} \right]$$

$$= -\sum_i \mathbf{p}_i \frac{p_\xi}{Q} \neq \mathbf{0}.$$

Note that we in our test we have enforced net zero total momentum at the beginning of each MD simulation, but non-conservative forces and slight numerical errors may still accumulate over MD trajectory. This non-zero momentum drift will induce non-zero COM drift over time as the MD simulations progress. Models failed to interact with heat bath correctly may not reproduce correct thermodynamic ensembles and therefore yield COM drift and incorrect partitions of kinetic and potential energies, as kinetic energies might be taken largely by COM velocity.

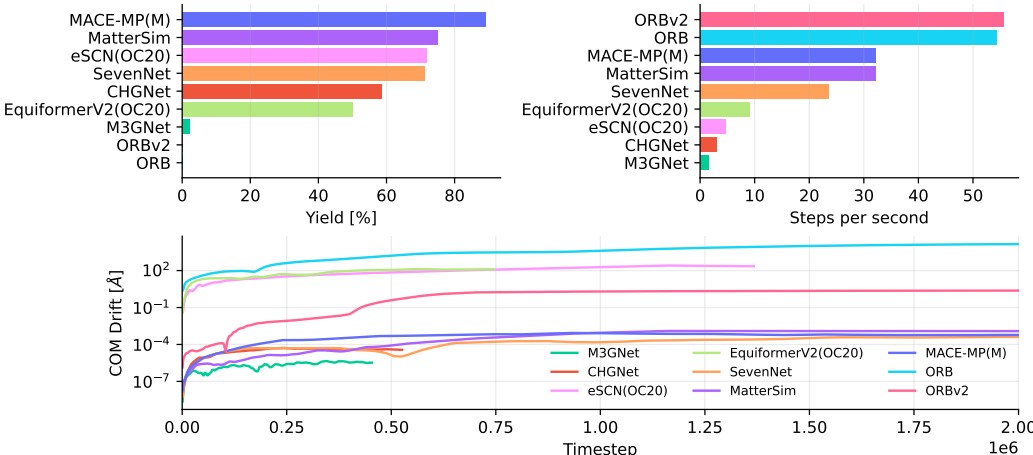

Figure S4: Hydrogen combustion. (Top left) The final reaction yield at the last MD step. (Top right) MD runtime speed measured in steps per second using single NVIDIA A100 GPU. `cuEquivariance` kernel was disabled for MACE-MP(M). (Bottom) The center-of-mass (COM) drift displacement during MD trajectory.

## A.8    Vacancy formation and migration in elemental solids

The benchmarking workflow included geometry optimization of pristine crystals, optimization of defective structure endpoints, and followed by climbing image nudged elastic band (CI-NEB) calculations [64] to identify transition states and determine vacancy migration barriers. Five intermediate images for NEB calculations were generated using the improved image-dependent pair potential (IDPP) method [65]. 57 FCC and 57 HCP crystals from Angsten et al. [31] consists of metallic, metalloid, and noble gas elements.

We define *path asymmetry* by calculating the mean difference between the left and right wings of normalized NEB profile $\epsilon(x) = \frac{E^{\text{ML}}(x)}{E^{\text{PBE}}_{\text{vm}}}$ with respect to the middle point $x = 0.5$:

$$\text{path asymmetry} = 2 \int_0^{0.5} |\epsilon(0.5 - x) - \epsilon(0.5 + x)| \, dx \qquad \text{(S8)}$$

.

*Barrier asymmetry* is defined as the ratio of reaction energy to forward barrier height:

$$\text{barrier asymmetry} = \frac{\Delta E}{E_{\text{forward}}} = \frac{E_f - E_i}{E_{\text{TS}} - E_i} \qquad \text{(S9)}$$

, where $E_i, E_f$ are energies of initial and final endpoints, and $E_{\text{TS}}$ is the transition state energy.

Figure S5 demonstrates the distribution of *barrier asymmetry* (eq. (S9)) of the vacancy migrations in elemental FCC and HCP crystals. We found that the compliance to symmetry is not strongly correlated with the equivariance and non-equivariance of the underlying MLIPs. MACE-MP(M) and MatterSim produce symmetric pathways. In contrast, ORBv2 and SevenNet tend to have asymmetric migration pathways, possibly due to more corrugated PES with multiple local minima where relaxation trajectories converge to. This might unintentionally lead to more undesirable behaviors and broken symmetries for sophisticated PES and diverse chemistry.

## A.9    Details on robustness under distribution shifts

Descriptors consist of two concatenated components, $X_i^R$ and $X_i^B$, that describe each central atom $i$'s radial distances to its $k$-nearest neighbors and its bond angles, respectively. $X_i^R$ is a vector of length $k$,

$$\left[ \frac{w(r_{i1})}{r_{i1}} \quad \dots \quad \frac{w(r_{ik})}{r_{ik}} \right]^T, \quad r_{ij} \leq r_{i(j+1)}, \qquad \text{(S10)}$$

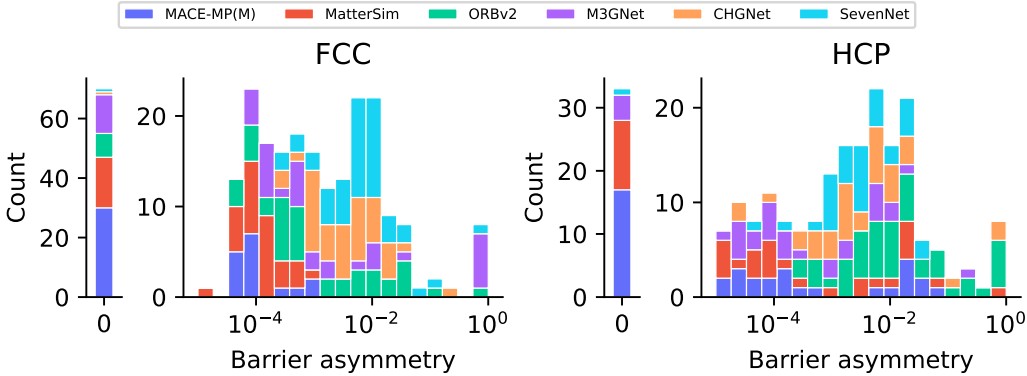

Figure S5: Absolute barrier asymmetry of vacancy migration in FCC and HCP elemental crystals. Compliance to symmetry is not correlated with the (non-)equivariance of the underlying MLIPs. Non-equivariant MLIPs: ORBv2, MatterSim, CHGNet. Equivariant MLIPs: MACE-MP(M), SevenNet.

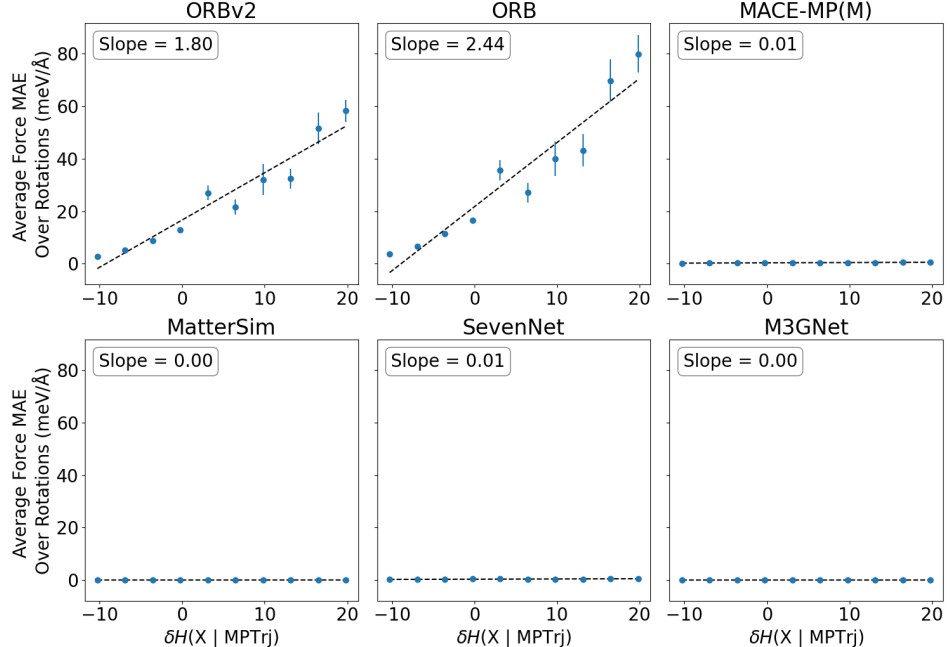

Figure S6: **Rotational equivariance versus differential entropy.** We calculate the mean absolute error (MAE) between each model's predicted forces and the forces predicted for rotated structures after transforming them back to the original reference frame. We compare this to the bin averages of differential entropy and report 95% confidence interval error bars for 10 bins from low to high differential entropy. Perfect rotational equivariance corresponds to a constant MAE of 0.0. Architectures without explicit rotational equivariance struggle to adhere to rotational equivariance with structures farther from the training distribution.

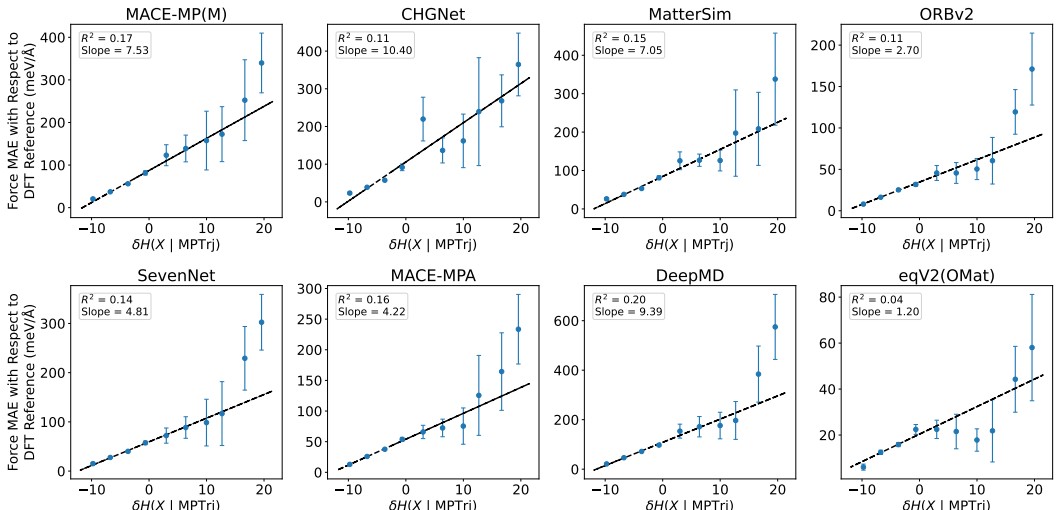

Figure S7: **Force MAE with respect to DFT versus differential entropy.** We compare the force MAE with respect to reference DFT values to the differential entropy of a random subset of 500 MPTrj structures. We report bin averages and 95% confidence interval error bars. Lines of best fit are provided. All models tend to predict forces less accurately when structures are more surprising.

where $1 \leq j \leq k$ due to the $k$-nearest neighbors approach, $r_{ij}$ is the distance between $i$ and another atom $j$, and $w(r)$ is a smooth cutoff function:

$$w(r) = \begin{cases} \left[ 1 - \left( \frac{r}{r_c} \right)^2 \right]^2, & 0 \leq r \leq r_c \\ 0, & r > r_c. \end{cases} \tag{S11}$$

To retain information not only about radial distances but also about bond angles, we use $X_{ijl}^B$ given by

$$\mathbf{X}_{ijl}^B = \frac{\sqrt{w(r_{ij})w(r_{il})}}{r_{jl}} \tag{S12}$$

which describes each neighbor $l$ of atom $j$ in the neighborhood of $i$. We represent the per-neighbor basis as

$$\mathbf{X}_{ij}^B = \left( X_{ij1}^B, \ldots, X_{ijk}^B \right), X_{ij1}^B \geq \cdots \geq X_{ij(k-1)}^B. \tag{S13}$$

For each atom, the bond angle descriptor then becomes

$$\mathbf{X}_i^B = \frac{1}{k} \sum_j \mathbf{X}_{ij}^B. \tag{S14}$$

To represent the training data, we compute these descriptors for every structure in each dataset. We hold out a subset of 50 structures in each dataset and estimate the Shannon information entropy of the remaining data using a kernel density estimate,

$$\mathcal{H}(\{\mathbf{X}\}) = -\frac{1}{n} \sum_{i=1}^{n} \log \left[ \frac{1}{n} \sum_{j=1}^{n} K_h(\mathbf{X}_i, \mathbf{X}_j) \right], \tag{S15}$$

where we use a Gaussian kernel:

$$K_h(\mathbf{X}_i, \mathbf{X}_j) = \exp\left(-\frac{\|\mathbf{X}_i - \mathbf{X}_j\|^2}{2h^2}\right). \tag{S16}$$

The bandwidth $h$ was selected according to the default provided by QUESTS [25], which was chosen to rescale the metric space of $\mathbf{X}$ according to the distance between two FCC environments with a 1% strain. To quantify the surprise of a data point $\mathbf{Y}$ compared to the existing observations $\{\mathbf{X_i}\}$, we define the differential entropy $\delta\mathcal{H}$ as

$$\delta\mathcal{H}(\mathbf{Y}|\{\mathbf{X}\}) = -\log\left[\frac{1}{n}\sum_{j=1}^{n} K_h(\mathbf{X}_i, \mathbf{X}_j)\right]. \tag{S17}$$

Figure S7 shows a correlation between force MAE and differential entropy for each model trained on MPTrj, indicating that the differential entropy is a reasonable measure of distribution shifts for MLIPs. Although these models perform well on in-distribution data, their error on force predictions increases as structures become more surprising, indicating a potential weakness in ability to generalize to out-of-distribution structures.

## A.10 Extended case studies

### A.10.1 CO$_2$ adsorption in metal-organic frameworks (MOFs)

Direct air capture (DAC) targets the removal of CO$_2$ directly from ambient air (about 400 ppm), and is increasingly recognized as indispensable for achieving net-negative greenhouse-gas emissions[66]. In practical, DAC technologies mainly rely on aqueous KOH slurries or amine-based absorbents whose chemisorptive binding affords the requisite affinity but imposes large thermal regeneration cost and chemical degradation. MOFs offer a promising physisorptive alternative for CO$_2$ capture. MOFs possess exceptionally high porosity and tunable structures that allow precise incorporation of functional groups such as open metal sites, diamines, thereby enhancing the affinity for CO$_2$ within their pore to levels suitable for DAC applications. Furthermore, the combination of framework rigidity, high surface area, and chemical stability positions MOFs as highly attractive candidates for durable, high-performance sorbents capable of operating under under relatively mild regeneration conditions.

In this case study, we curated 20 MOFs with experimentally reported $Q_{st}$ values spanning three technologically relevant adsorption regimes and evaluated how accurately MLIPs classify them into correct categories: (1) General ($Q_{st} < 35\,kJ/mol$), (2) post-combustion flue gas ($35\,kJ/mol < Q_{st} < 50\,kJ/mol$), and (3) DAC ($50\,kJ/mol < Q_{st} < 100\,kJ/mol$).

The general adsorption class includes seven representative MOFs such as MOF-5, HKUST-1, UiO-66, ZIF-8, MIL-177, MIL-53-Al, and MOF-74-Fe. They exhibit relatively low CO$_2$ affinities and have not been prominently reported for CO$_2$ capture applications. The post-combustion flue gas class corresponds to MOFs reported for capturing CO$_2$ from power-plant exhaust, where the partial pressure of CO$_2$ is much higher than in ambient air, thus requiring moderate adsorption strengths. This category includes CALF-20, Al-PyMOF, UTSA-16, MUF-16, and ZnH-MFU-4l. Finally, the DAC class comprises MOFs capable of capturing CO$_2$ at extremely low partial pressures, demanding high binding affinities. DAC-relevant materials considered here include SIFSIX-3-Cu, NbOFFIVE-1-Ni, TIFSIX-3-Ni, SIFSIX-18-Ni-beta, en-Mg2(dobpdc), and CFA-1-OH-Zn, and SGU-29.

Table S3: Experimentally reported CO$_2$ Q$_{st}$ values used in MLIP-arena, including three categories: DAC, post-combustion flue gas, normal MOFs.

| Common name | CO$_2$ Q$_{st}$ (kJ/mol) | Category | Reference |
|---|---|---|---|
| SIFSIX-3-Cu | 54 | DAC | [67] |
| NbOFFIVE-1-Ni | 54 | DAC | [68] |
| TIFSIX-3-Ni | 49 | DAC | [68] |
| SIFSIX-18-Ni-$\beta$ | 52 | DAC | [68] |
| en-Mg$_2$(dobpdc) | 50 | DAC | [69] |
| CFA-1-OH-Zn | 42 (71 in max) | DAC | [70] |
| SGU-29 | 51.3 | DAC | [71] |
| CALF-20 | 39 | Post-combustion flue gas | [72] |
| Al-PyrMOF | 28 | Post-combustion flue gas | [73] |
| UTSA-16 | 39.7 | Post-combustion flue gas | [74] |
| MUF-16 | 32.3 | Post-combustion flue gas | [75] |
| MIL-120-Al-AP | 41 | Post-combustion flue gas | [76] |
| ZnH-MFU-4l | 20 (93 in high T) | Post-combustion flue gas | [77] |
| Fe-MOF74 | 33.2 | General | [78] |
| MIL-53-Al | 26.3 | General | [79] |
| HKUST-1 | 29 | General | [80] |
| MOF-5 | 15 | General | [81] |
| UiO-66 | 28.6 | General | [82] |
| ZIF-8 | 27 | General | [83] |
| MIL-177 | 14 | General | [84] |

The heat of adsorption is calculated from the statistical average of interaction energies $E_{int}$ using Widom insertion method [32, 85]. The interaction energy $E_{int}$ is determined by energy difference between gas-inserted MOF and individual gas and MOF system:

$$E_{int} = E_{MOF+gas} - E_{MOF} - E_{gas}. \tag{S18}$$

The heat of adsorption is then determined from ensemble average:

$$Q_{st} = -\frac{\langle E_{int} e^{-\beta E_{int}} \rangle}{\langle e^{-\beta E_{int}} \rangle} + k_B T,$$ (S19)

where $\beta = (k_B T)^{-1}$, $k_B$ is Boltzmann constant, and $T$ is temperature.

All MLIP models are used in combination with D3 Becke-Johnson dispersion correction [86] with cutoff of 40 Bohr radius. Initial MOF structures were first alternately optimized with fixed and relaxed cell protocols until the final maximum atomic force is smaller than $0.05\,\mathrm{eV/\mathring{A}}$. The Widom insertion of $CO_2$ was then performed at $300\,\mathrm{K}$ for three rounds for each MOF, with 5,000 insertion trials in each round. The grid spacing between gas insertion points was set at $0.15\,\mathring{A}$.

Figure S8 shows distribution of predicted heat of $CO_2$ adsorption and average misclassification margin across 20 MOFs. The misclasification margin is defined as the distance between misclassified point to the closest decision boundary. Our results show that MatterSim is the strong MOF classifiers with misclassification margin of $11.30\,\mathrm{kJ/mol}$ and misclassification count only 4, while MACE-MP(M) and SevenNet have severe overestimation, possibly due to short-range PES holes.

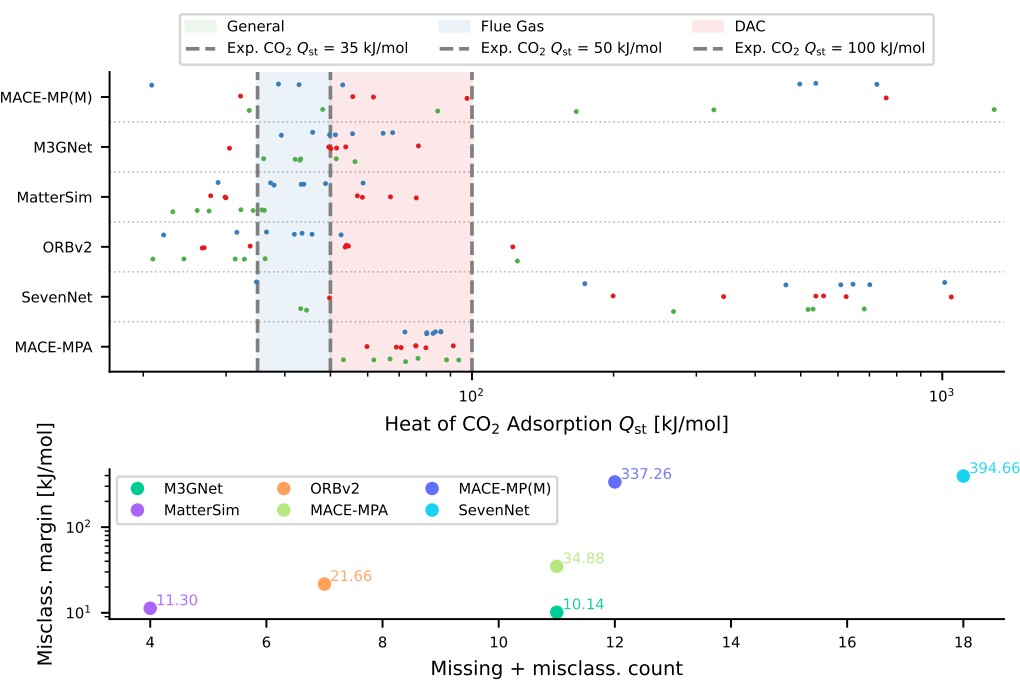

Figure S8: **Classification of MOFs based on predicted heat of $CO_2$ adsorption.** (Top) Three classes of MOFs based on experimental $CO_2$ $Q_{st}$ measurements: (1) general (green area and points), (2) flue gas (blue area and blue points), and (3) DAC (red area and points). The perfect classifiers should predict $Q_{st}$ of $CO_2$ in the corresponding regions. (Bottom) Mean misclassification margin and count of misclassified and missing MOFs.

### A.10.2 Dynamical stability of 2D materials

Two-dimensional materials are vital to emerging technologies due to their exceptional physical properties and chemical tunability. To evaluate the ability of MLIPs to predict dynamical stability, we randomly selected 505 monolayers from the C2DB database [34, 87] and computed elastic tensors and phonon band structures using Pymatgen [88] and Phonopy [89, 90]. Following the C2DB protocol, a material is labeled dynamically stable if both the elastic tensor eigenvalues and lowest phonon frequencies are non-negative (specifically, both values should be greater than $-10^{-7}$ to be labeled as stable).

F1 scores (fig. S9) indicate that MACE-MP(M), MACE-MPA, and MatterSim perform best, with macro F1 scores of 0.420, 0.412, and 0.411, respectively. In contrast, CHGNet, ORBv2, and ALIGNN perform significantly worse, with macro F1 scores below 0.30 and stable F1 scores of 0. All models show higher F1 scores for the unstable class—e.g., 0.596 for MACE-MP(M) vs. 0.245 for stable—highlighting a bias toward detecting instability. Confusion matrices (fig. S10) confirm this trend: most models heavily misclassify stable materials as unstable, with CHGNet, ORBv2, and ALIGNN failing to identify any stable structures. These findings reflect current limitations in MLIP generalization to vibrational stability and emphasize the need for improved training strategies that target phonon-related properties.

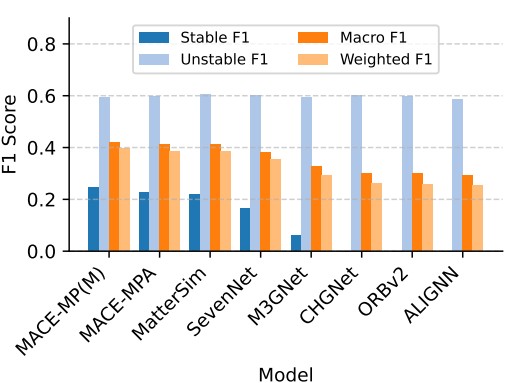

Figure S9: F1 scores of dynamical stability classification for 2D materials from C2DB database.

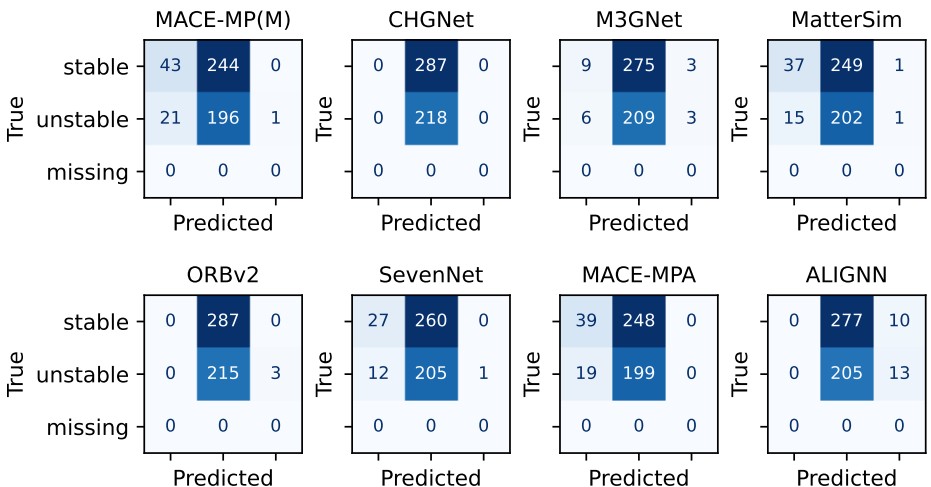

Figure S10: Confusion matrices of dynamical stability classification for 2D materials from C2DB database.

### A.10.3 Second-order dynamical phase transition in perovskite

Perovskites are a versatile class of materials exhibiting diverse properties, including ferroelectricity, magnetoresistance, ionic conductivity, piezoelectricity, and superconductivity. Barium zirconate ($BaZrO_3$, BZO) has been predicted and observed to have a second-order phase transition due to dynamical instability in the cubic polymorph [33, 37]. In Figure S11, we probe the anharmonic PES of different MLIPs along the octahedral-tilting phonon mode with different unit cell lattice constants. Energy differences are calculated with respect to the undeformed structures at the respective lattice constants. We observe Landau-like second-order phase transition from quartic to quadratic polynomials in MACE-MP(M), MatterSim, CHGNet, and SevenNet. M3GNet remains in quadratic PES across all structures with close degeneracies. ORBv2 has an asymmetrical PES and multiple energy crossings.

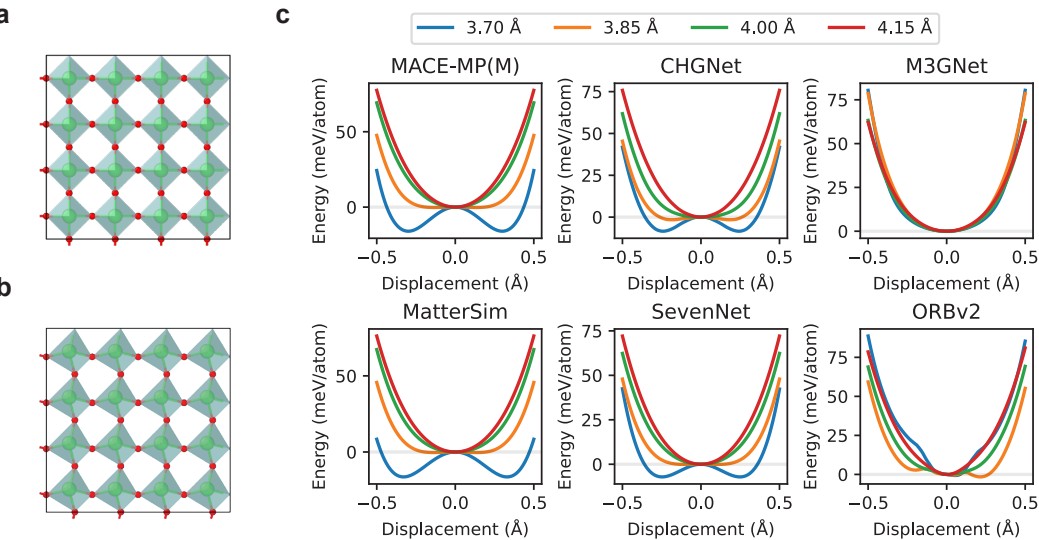

Figure S11: Landau-like second-order phase transition of octahedral-tilting mode in $BaZrO_3$ (BZO). (a) Undeformed $4 \times 4 \times 4$ supercell of BZO with cubic unit cell lattice constant of $4\,\text{Å}$. (b) R-tilt phonon mode with maximum displacement of $0.5\,\text{Å}$. Ba atoms are transparent for better visualization. (c) Transitional behavior from quadratic to quartic Landau-like potential energy landscape as a function of largest modal displacement for different lattice constants from $3.70\,\text{Å}$ to $4.15\,\text{Å}$.

# B Supported models

Table S4: List of first-class supported open-source, open-weight models in MLIP Arena. Custom models could be incorporated through convenient class inherited from `ASE Calculator`.

| Model | Prediction[1] | NVT | NPT | Training Set[2] | Code | Reference | License | Checkpoint | First Release |
|---|---|---|---|---|---|---|---|---|---|
| MACE-MP(M) | EFS | ✓ | ✓ | MPTrj | GitHub | Batatia et al. [27] | MIT | `2023-12-03-mace-128-L1_epoch-199.model` | 2023-12-29 |
| CHGNet | EFSM | ✓ | ✓ | MPTrj | GitHub | Deng et al. [26] | BSD-3-Clause | `v0.3.0` | 2023-02-28 |
| M3GNet | EFS | ✓ | ✓ | MPF | GitHub | Chen and Ong [28] | BSD-3-Clause | `M3GNet-MP-2021.2.8-PES` | 2022-02-05 |
| MatterSim | EFS | ✓ | ✓ | MPTrj, Alex, Proprietary | GitHub | Yang et al. [29] | MIT | `MatterSim-v1.0.0-5M.pth` | 2024-05-10 |
| ORB | EFS | ✓ | ✓ | MPTrj, Alex | GitHub | N/A | Apache-2.0 | `orbff-v1-20240827.ckpt` | 2024-09-03 |
| ORBv2 | EFS | ✓ | ✓ | MPTrj, Alex | GitHub | Neumann et al. [23] | Apache-2.0 | `orb-v2-20241011.ckpt` | 2024-10-15 |
| SevenNet | EFS | ✓ | ✓ | MPTrj | GitHub | Park et al. [30] | GPL-3.0 | `7net-0` | 2024-07-11 |
| eqV2(OMat) | EFS | ✓ | ✗ | OMat, MPTrj, Alex | GitHub | Barroso-Luque et al. [5] | Apache-2.0* | `eqV2_86M_omat_mp_salex.pt` | 2024-10-18 |
| eSEN | EFS | ✓ | ✓ | OMat, MPTrj, Alex | GitHub | Fu et al. [51] | Apache-2.0* | `esen_30m_oam.pt` | 2025-04-14 |
| EquiformerV2(OC22) | EF | ✓ | ✗ | OC22 | GitHub | Liao et al. [22] | Apache-2.0 | `EquiformerV2-lE4-lF100-S2EFS-OC22` | 2023-06-21 |
| EquiformerV2(OC20) | EF | ✓ | ✗ | OC20 | GitHub | Liao et al. [22] | Apache-2.0 | `EquiformerV2-31M-S2EF-OC20-All+MD` | 2023-06-21 |
| eSCN(OC20) | EF | ✓ | ✗ | OC20 | GitHub | Passaro and Zitnick [91] | Apache-2.0 | `eSCN-L6-M3-Lay20-S2EF-OC20-All+MD` | 2023-02-07 |
| DeepMD | EFS | ✓ | ✓ | MPTrj | GitHub | Zhang et al. [92] | GNU LGPLv3.0 | `dp0808c_v024mixu.pth` | 2024-10-09 |
| ALIGNN | EFS | ✓ | ✓ | MP22 | GitHub | Choudhary and DeCost [93] | NIST | `2024.5.27` | 2021-11-15 |

[1] E: energy, F: force, S: stress, M: magmom.
[2] MPTrj: Materials Project GGA-PBE relaxation trajectories, Alex: Alexandria GGA-PBE dataset [4], OMat: Open Materials dataset [5], MP22: Materials Project 2022, MPF: MPF.2021.2.8: Materials Project snapshot curated to train M3GNet [28]. OC20, OC22: Open Catalyst Project [94, 95].
*Modified Apache-2.0 (Meta)

# C   Additional DFT reference benchmarks

**Bulk modulus from equation of state (EOS) calculations.**   In the vacancy migration task (appendix A.8), the geometry optimization of each pristine structure is then followed by an EOS fit to compare with GGA-PBE data from Angsten et al. [31]. Figure S12 shows that most of the model can capture the trend up to $400\,\mathrm{GPa}$ well, with serious underestimation on a few FCC and several HCP structures.

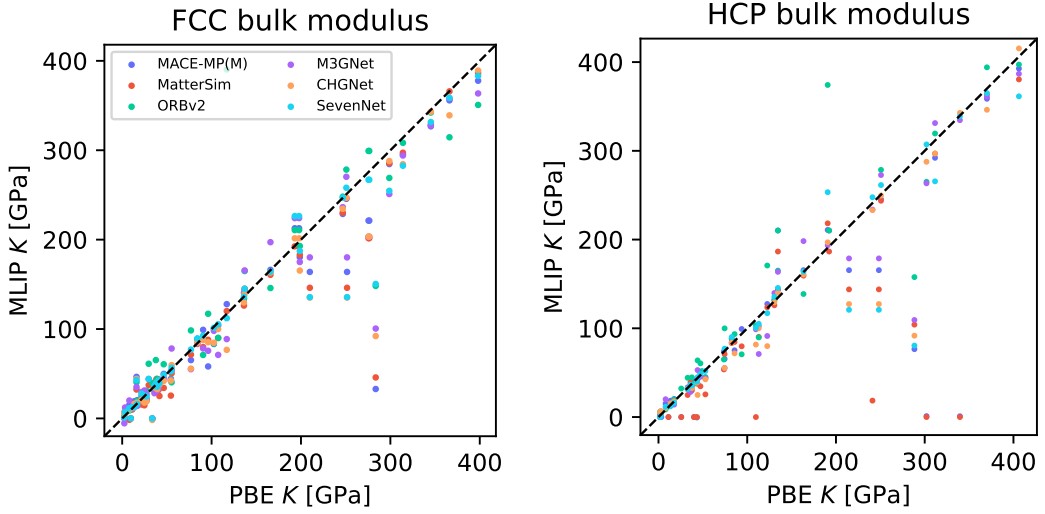

Figure S12: Bulk modulus of FCC and HCP elemental solids compared with GGA-PBE calculations [31].

Table S5: Bulk modulus of FCC elemental crystals. nNA denotes the number of missing predictions out of 57 entries except for noble gases.

| model | MAE (GPa) | MAPE (%) | nNA |
|---|---|---|---|
| MACE-MP(M) | 18.878 | 28.7 | 2 |
| MatterSim | 19.142 | 28.1 | 1 |
| ORBv2 | 32.583 | 31.5 | 1 |
| M3GNet | 21.867 | 37.0 | 4 |
| CHGNet | 19.815 | 25.5 | 6 |
| SevenNet | 14.500 | 21.1 | 3 |

Table S6: Bulk modulus of HCP elemental crystals. nNA denotes the number of missing predictions out of 57 entries except for noble gases.

| model | MAE (GPa) | MAPE (%) | nNA |
|---|---|---|---|
| MACE-MP(M) | 35.969 | 36.3 | 5 |
| MatterSim | 45.865 | 35.5 | 5 |
| ORBv2 | 41.116 | 36.4 | 4 |
| M3GNet | 21.321 | 22.0 | 16 |
| CHGNet | 21.484 | 26.3 | 16 |
| SevenNet | 21.925 | 17.0 | 15 |

