# OpenReview forum: "MLIP Arena: Advancing Fairness and Transparency in Machine Learning Interatomic Potentials via an Open, Accessible Benchmark Platform"
_NeurIPS.cc/2025/Datasets_and_Benchmarks_Track — NeurIPS 2025 Datasets and Benchmarks Track spotlight_

### Official Review · Reviewer_nH4R · 2025-06-19

**Rating:** 5
**Confidence:** 4

**Summary:**

This paper presents MLIP Arena, an benchmarking platform that evaluates machine learning interatomic potentials (MLIPs) through four key dimensions: asymptotic behavior under extreme conditions, stability in molecular dynamics simulations, robustness to distribution shifts, and predictive accuracy for thermodynamic properties. The work makes a valuable contribution by moving beyond traditional error metrics to assess physical consistency and practical utility. The authors demonstrate that current MLIPs exhibit significant limitations in chemical reactivity and out-of-distribution generalization, despite performing well on standard benchmarks.

The study is technically sound and well-executed, with thorough experiments and clear presentation of results. The open-source implementation and public leaderboard enhance reproducibility and community adoption. However, the paper could be strengthened by including more diverse chemical reactions in the reactivity tests and discussing potential dataset biases. The computational requirements for running these benchmarks could also be clarified.

**Dataset Code Accessibility:**

Yes

**Ethical Considerations:**

No, there are no or only very minor ethics concerns

**Limitations Weaknesses:**

The paper relies heavily on specialized abbreviations (PES/PEC/EOS) without sufficient explanation, creating barriers for non-specialist readers.

The flow between sections feels disjointed, particularly the jump from Introduction to technical details. A bridging paragraph summarizing the four evaluation dimensions before Section 2.1 would improve coherence.

The discussion of model failures (e.g., ORBv2's hydrogen combustion performance) remains surface-level. Deeper architectural analysis (e.g., how non-conservative force prediction conflicts with physical constraints) would strengthen insights.

**Strengths Contributions:**

The paper moves beyond conventional error-based metrics by introducing physics-aware evaluation criteria (e.g., asymptotic behavior, conservation laws, and reactivity), providing a more holistic assessment of MLIPs' real-world applicability.

The open-source platform and public leaderboard enable standardized, transparent comparisons of MLIPs, fostering community-wide progress in the field. The comprehensive experiments reveal critical limitations of current models, guiding future development.

---

> ### Author Rebuttal · Authors · 2025-07-31
>
> **W1:**
> Thank you for the suggestion. We will add a paragraph in the introduction section to increase the accessibility for the non-specialist practitioners.
>
> **W2:**
> We appreciate the suggestion. We will add a transition paragraph to summarize and discuss the four benchmark categories before section 2.1.
>
> **W3:**
> We appreciate the comment. Direct force models are referred to the models with separate force readout head instead of taking the analytic gradient of the energy prediction. In section A.7 (line 736), we derive a complementary proof on the momentum non-conservativeness of a model with enforced net zero force under Nosé-Hoover thermostat integration. We will add more discussion in the main text and refer the readers to the derivation in the appendix.

---

### Official Review · Reviewer_HJB4 · 2025-06-24

**Rating:** 6
**Confidence:** 2

**Summary:**

The benchmark is an established, highly-relevant framework for testing ML-based interatomic potentials (MLIPs). Authors argue that traditional benchmarks that rely on static error-based metrics against DFT references are insufficient to estimate practical utility of MLIPs in real-world applications. It offers evaluating the ability of MLIPs to capture correct dynamic interactions of atomistic systems by testing how they satisfy physical laws and symmetries, which is extremely sound for real-world practitioners. The benchmark is open, well-documented, and includes an online leaderboard.

**Dataset Code Accessibility:**

Yes

**Dataset Code Comments:**

The code is open-sourced, clean and well-documented.

**Ethical Comments:**

The benchmark is technical.

**Ethical Considerations:**

No, there are no or only very minor ethics concerns

**Final Justification:**

I believe the submission is very valuable and as such keep my score to highlight it.

**Limitations Weaknesses:**

- My main concern boils down to the statistical interpretation of the results. For example, in Table 1, the standard deviations for several metrics are large relative to the differences in the mean values between models. For the 'Spearman's coefficient: compression', the top three models have nearly identical means and overlapping standard deviations. Furthermore, can a meaningful conclusion about the superiority of one model over another be drawn from this specific task? A discussion on the statistical significance of the differences would strengthen the paper's conclusions.
- The paper notes that autoregressive benchmarks are more computationally expensive than static DFT predictions. While the work is positioned as a benchmark, a more explicit discussion and quantification of the computational cost (e.g., total GPU hours to run one model through the entire benchmark) would be a valuable addition. This would help potential users prepare the resources required to evaluate their own models and understand the trade-offs.
- The benchmark includes hydrogen combustion as its primary test for chemical reactivity. While this is a challenging out-of-distribution task, it represents only a single type of reaction (gas-phase combustion). This makes it difficult to generalize a model's overall performance on reactivity.
- A significant portion of the benchmarks, including the EOS, vacancy migration, 2D materials stability, and perovskite phase transition tests, are focused on crystalline solids. While important, this focus may not adequately evaluate the performance of MLIPs on other domains: liquids, polymers, complex biomolecules, etc.

**Strengths Contributions:**

- Well documented and highly relevant becnhmark to estimate MLIPs application to simulating real-world scenarios such as material generation.
- The evaluation is not limited to DFT-based references and highlights applicability of a model to out-of-distribution tasks.
- The framework includes 4 distinct evaluation categories: (1) asymptotic behaviors, (2) stability/reactivity under extreme conditions, (3) robustness to distribution shifts, and (4) thermodynamic properties. Each category targets different failure modes of current MLIPs.
    - asymptotic behaviors: crucial for ensuring models behave physically at extreme separations or compressions.
    - stability and reactivity: directly tests the primary use case of MLIPs (material generation).
    - robustness to distribution shifts which tests how models maintain energy conservation and rotational equivariance subjected to unseen scenarios.
    - thermodynamic properties: complex calculations that are highly relevant for materials science, which is one of primary use cases of MLIPs.
- The frameworks comes with online leaderboard which promotes transparency.
- All the major foundational MLIPs aretested, providing a comprehensive comparison. There are plenty of tested models, including those with different properties (e.g., equivariant vs. non-equivariant) and training datasets, which is valuable for guiding design choices in the future.

---

> ### Author Rebuttal · Authors · 2025-07-31
>
> **W1:**
> We appreciate the comment and will strengthen the discussion on the statistical significance during revision. We have also answered the similar point by reviewer Y322 W1. As many models start to saturate the metrics, the community needs benchmarks that are able to cover comprehensive, orthogonal metrics to reliably and fairly distinguish the performance. Prior work on Matbench Discovery [1] is an example of focusing solely on energy related metrics (energy to the convex hull and its regression and classification). Although multiple metrics are available on the leaderboard, most of them are highly interdependent and ignore the fact that MLIP should be used as the potential that also needs to perform well on force, stress, stability, symmetry compliance and more we explore in MLIP Arena.
>
> [1] Riebesell, Janosh, et al. "A framework to evaluate machine learning crystal stability predictions." Nature Machine Intelligence 7.6 (2025): 836-847.
>
> **W2:**
> We appreciate the suggestion. While unfortunately we didn’t record the GPU/CPU runtime cost for all benchmarks except for MD stability tests, during the revision we will work on the code infrastructure to record the runtime statistics for all tests, and will provide in the appendix where appropriate. We also consider reviewer Y322’s suggestion to add the runtime performance of classical forcefields (LJ, ZBL) to the MD stability tests.
>
> **W3:**
> Thank you for the suggestion. As the benchmark continues to grow, in the future we will include more reaction types. To constrain our discussion in the current form of text, we will change the subsection title “chemical reactivity” at line 139 to “hydrogen combustion.”
>
> **W4:**
> Thank you for the valuable suggestion. We will continue work on extending the benchmark to small molecules, for example torsion scan (e.g. TorsionNet 500 [1]), radius of gyration or radial distribution function over MD simulations to reach broader audiences.
>
> [1] Rai, Brajesh K., et al. "Torsionnet: A deep neural network to rapidly predict small-molecule torsional energy profiles with the accuracy of quantum mechanics." Journal of Chemical Information and Modeling 62.4 (2022): 785-800.

---

### Official Review · Reviewer_Y322 · 2025-06-25

**Rating:** 5
**Confidence:** 4

**Summary:**

This paper introduces MLIP Arena, a new benchmarking platform for machine learning interatomic potentials (MLIPs). Unlike previous benchmarks, MLIP Arena emphasizes physical realism, chemical reactivity, stability under extreme conditions, and predictive accuracy for thermodynamic properties. The benchmark results reveal key limitations in current MLIPs and offer a reproducible, physics-informed framework to guide the development of more accurate and efficient force fields. Users can explore MLIP Arena through a Python package and an online leaderboard.

**Additional Feedback:**

* The color scheme in Figure 2 is not defined—please include a legend or explanation.

* Provide details on each MLIP model used—e.g., whether they are direct force or gradient-based—in Table S4 or the main text.

* In Equation S1, why not directly use ∣curl F∣ as the metric? Please clarify.

* In Section 2.2 (stability under extreme conditions), the simulations ramp temperature from 300 K to 3000 K and pressure from 0 to 500 GPa over 10 ps. Please justify whether 10 ps exceeds the relaxation time of the systems involved; otherwise, the MD simulations may not represent equilibrium behavior. Also, including results from classical force fields in Figure 3 would add valuable context.

* Lines 130–136 in Section 2.2 could be clarified by directly presenting the computational runtime of each MLIP model.

* The ability of MLIPs to capture short-range stiffness likely depends on whether the training data includes such scenarios. Please justify whether the training data appropriately covers this regime and whether accurate modeling of short-range stiffness is expected from the MLIPs used.

* Most equations in the manuscript lack proper punctuation (commas or periods), which even comes the text just after the equations. This disrupts the flow and readability. Please revise accordingly for stylistic consistency with scientific writing standards.

**Dataset Code Accessibility:**

Partly

**Dataset Code Comments:**

* As noted above, the paper lacks API usage documentation. This information is crucial for a benchmark/dataset paper. The GitHub documentation is insufficient, making it difficult for newcomers to adopt MLIP Arena. Consider adding a Jupyter notebook with example use cases or improving inline documentation.

* The Quickstart example on GitHub fails when MLIP Arena is installed via PyPI (at least my environment created using mamba). Please fix this issue or clarify the installation instructions.

**Ethical Comments:**

n.a

**Ethical Considerations:**

No, there are no or only very minor ethics concerns

**Final Justification:**

Thank you to the authors for their diligent efforts in addressing my comments. I feel that most of my concerns have been appropriately handled, especially when the responses are properly reflected in the revisions. Finally, I encourage the authors to make additional efforts in documenting the codebase (e.g., README) to further enhance the user-friendliness of the dataset.

**Limitations Weaknesses:**

Major:

* W1: Several proposed metrics (e.g., Table 1 and Table S2) do not effectively distinguish MLIP performance. In particular, the results in Table 1 are all within 1-sigma, making it difficult to interpret differences. In contrast, Figure 2 is much more informative. Consider introducing metrics that can better capture such distinctions.

* W2: There is a lack of detailed information about the evaluated MLIPs, including their force/energy error benchmarks and the training datasets used. This omission hinders any meaningful discussion about the origins of the observed failure modes.

* W3: Figure S1 is difficult to interpret without a reference curve obtained from DFT or high-accuracy methods such as coupled cluster.

* W4: The explanation of benchmark tasks needs to be more comprehensive. For instance, in Section 2.3, it is unclear which materials are being tested. If C–H bonds are involved, a 1 fs time step may be too large.

* W5: The explanation of the smoothness metric (Lines 633–635) is inconsistent: it suggests the optimal value can be above unity, yet the ranking is based on deviation from unity, which is misleading. Consider revising the ranking approach or removing it altogether if it cannot be rigorously justified.

* W6: The paper lacks a description of the MLIP Arena API. For a benchmark/dataset paper, such guidance is essential for usability. Furthermore, the GitHub documentation is insufficient, making it difficult for new users to adopt the platform without significant effort. Providing Jupyter notebooks or better documentation would greatly improve usability.

Minor:

* W7: Include a summary table of the proposed metrics and tasks to improve clarity, potentially replacing or shortening the author contributions and acknowledgments sections.

* W8: Consider including MLIP models already validated in real-world computational chemistry applications, such as NequIP and Allegro.

* W9: Regarding chemical reactivity: since even DFT struggles with can offer true dynamic bond-breaking processes due to the dynamic nature, it is unclear what value the proposed tests provide without a definitive reference. Please clarify the intended insight from these comparisons. In particular, what is the definition of "target" in Figure S3? I could not its description, other than a short one in caption of Figure S3 as "experimental reaction ...". Clarify this part more either in main body or appendix.

**Strengths Contributions:**

* The codebase is well-prepared, available on GitHub and even published on PyPI, facilitating easy access for users.

* Pretrained MLIP model checkpoints are hosted on HuggingFace, enhancing reproducibility and ease of deployment.

* The paper proposes a variety of new benchmark tasks grounded in real-world applications, offering valuable insights into MLIP performance from both physical and chemical perspectives.

---

> ### Author Rebuttal · Authors · 2025-07-31
>
> **W1:**
> Thank you for the comment. Similar to the rapid saturation seen in the LLM space, foundation MLIPs often saturate benchmark metrics with overlapping standard deviations. Despite small differences within individual metrics, we adopt a multi-metric strategy with rank aggregation to distinguish performance more reliably (as reflected in the third column of Table 1). We also note that high-ranked metrics tend to show smaller standard deviations. We will present Figure 2 before Table 1 to improve clarity and consider using min/max values across structures instead of standard deviations as alternative performance indicators.
>
> **W2:**
> We thank the reviewer for the suggestion. Appendix B Table S4 lists model checkpoints and training sets. We will reference this in the main text and clarify that energy/force errors are dependent on the DFT reference and not central to this work, as all models are pretrained with their own reported errors on separate training sets.
>
> **W3:**
> In this work, we benchmark the geometric measures of diatomic dissociation curves that all models should satisfy, independent of DFT or CC references. Although comparisons with gold standard coupled cluster calculation are desirable, the diatomic curves predicted by many MLIPs already reveal serious failure modes that can be identified without DFT/CC references. We highlight a common oversight in MLIP development: focusing on energy/force regression in condensed phases does not guarantee correct atomic interactions or generalization to off-equilibrium scenarios.
>
> Most listed models, and many universal MLIPs, rely on the plane-wave DFT code VASP, which is known to be unreliable for isolated dimers. We attempted PBE curves for several elements and shared the results on our leaderboard; many others faced convergence issues. Even CCSD(T) can fail, especially for certain diatomics like N2, when using single-reference treatments [1].
>
> [1] Kats, Daniel, and Frederick R. Manby. "Communication: The distinguishable cluster approximation." The Journal of chemical physics 139.2 (2013).
>
> **W4:**
> We thank the reviewer for the suggestion. These experiments were performed on the subset of MPtrj. The timestep of 1 fs is frequently adopted in protein [1], inorganic material [2], and small molecule [3] MD simulations. For our MD stability benchmark, the timestep is set to 0.5 fs for any structure containing H atoms, as indicated by the MD task docstring in the repo. We will highlight these points in the main text during revision.
>
> [1] Sahil, Mohammad, Susmita Sarkar, and Jagannath Mondal. "Long-time-step molecular dynamics can retard simulation of protein-ligand recognition process." Biophysical Journal 122.5 (2023): 802-816.
>
> [2] Kaplan, Aaron D., et al. "A foundational potential energy surface dataset for materials." arXiv preprint arXiv:2503.04070 (2025).
>
> [3] Fu, Xiang, et al. "Learning smooth and expressive interatomic potentials for physical property prediction." arXiv preprint arXiv:2502.12147 (2025).
>
> **W5:**
> We thank the reviewer for the detailed review on supplementary information, and we apologize for the confusion. For the equation of state benchmark in the main (and energy-volume scan in the appendix), the definition of tortuosity is unambiguous and the optimal value is 1. Lines 633–635 only apply to the potential energy curves (PECs) of homonuclear diatomics, and we explain the consideration below:
>
> All the current foundation MLIPs are meant to fit to the ground state PES (to be more specific, lowest nonrelativistic adiabatic surface under Born-Oppenheimer approximation). This means the homonuclear diatomic curves have strictly a single well before the ground state PES smoothly approaches dissociation limit. As we constrain our discussion on homonuclear diatomics, any charge transfer will always lead to configurations with energy higher than ground state PES. Therefore, all the tortuosity should be equal to 1 under our definition.
>
> However in reality, some elements have d- or f- electrons with strong spin-orbit coupling or the nuclei might move at significantly high speed at non-negligible fraction of electrons, leading to the breakdown of BO approximation. When two BO surfaces are degenerate around the (conical) intersection, the complex electronic structure and hopping might result in a small energy hump with minute increase in PEC tortuosity. We will remove the tortuosity from the rank aggregation for the homonuclear diatomic during revision.
>
> Albeit these potential breakdowns, the notorious Cr-Cr dimers, even though previously predicted to have secondary energy local minimum (tortuosity > 1), has been recently demonstrated [1] that such spurious minimum does not exist and the proposed composite method (X2C+DMRG+SHCI) agrees quantitatively with experiment [2]. Therefore the current most correct tortuosity of Cr-Cr dissociation curve restores back to 1.
>
> [1] Larsson, H. R., Zhai, H., Umrigar, C. J., & Chan, G. K. L. (2022). The chromium dimer: closing a chapter of quantum chemistry. Journal of the American Chemical Society, 144(35), 15932-15937.
>
> [2] Casey, S. M., & Leopold, D. G. (1993). Negative ion photoelectron spectroscopy of chromium dimer. The Journal of Physical Chemistry, 97(4), 816-830.
>
> **W6:**
> Our benchmark scripts are in the `/benchmark` folder, provided as either Python files or notebooks depending on the task. We will add detailed instructions during the revision. Thank you for the feedback.
>
> **W7:**
> We have indicated in the Figure 2 caption in our original submission: “Color represents EOS curve of each crystal structure.”
>
> **W8:**
> Thank you for the suggestion. To our knowledge, there is no foundation checkpoint for NequIP or Allegro. SevenNet and eSEN are efficient derivatives of NequIP.
>
> **W9:**
> Thank you for the comment. Dashed lines appear in both top and bottom panels. In the top panel, they indicate the target temperature; in the bottom, they refer to experimental reaction enthalpies. We will clarify in both the caption and main text.
>
> ---
>
> ### **Dataset Code Comments**
>
> **D1+D2+D3:**
> Thank you for the feedback and apologies for the unclear instructions. We provide notebooks or `.py` files in the `/benchmark` folder and will add detailed task-specific instructions.
>
> The README states that the PyPI installation includes only the Prefect workflow, not pretrained models. To reproduce our environment and install compatible models, we recommend running `bash scripts/install.sh` from `/scripts`. Note that our Prefect workflow supports any ASE-compatible calculator, so users can choose Arena models or use custom MLIPs.
>
> ---
>
> ### **Additional Feedback**
>
> **A1:**
> The caption of Figure 2 specifies that each color represents a single crystal structure for clearer visualization.
>
> **A2:**
> Thank you. We will include this information in Table S4 and discuss the baseline models in the main text.
>
> **A3:**
> We examine a one-dimensional PES slice by projecting along the internuclear axis. For diatomics, force should be strictly along this axis with no perpendicular component.
>
> **A4:**
> Thank you for raising the concern. We limited MD simulations to 10 ps per trajectory due to computational constraints. While this duration may be shorter than some system relaxation times, the systems studied are amorphous mixtures, which typically exhibit short stress correlation times. Our setup approximates isotropic shock compression.
>
> As reference, [1] applies Nosé-Hoover Hugoniostat (NVHug and NPHug) [2] to increase system’s pressure 0 to 20 GPa and temperature to 4500K over the course of 0.5 ps, at the same scale of stress rate with our simulations. Earlier study [3] also investigated the Hugoniot states of MgO at the condition up to 1.2 TPa and 42,000 K. We will add these references. Thank you for raising this concern.
>
> [1] Wilkins, J. S., & Probert, M. I. (2024). An ab initio approach to the Hugoniot. The Journal of Chemical Physics, 161(14).
>
> [2] Ravelo, R., Holian, B. L., Germann, T. C., & Lomdahl, P. S. (2004). Constant-stress Hugoniostat method for following the dynamical evolution of shocked matter. Physical Review B—Condensed Matter and Materials Physics, 70(1), 014103.
>
> [3] Root, S., Shulenburger, L., Lemke, R. W., Dolan, D. H., Mattsson, T. R., & Desjarlais, M. P. (2015). Shock response and phase transitions of MgO at planetary impact conditions. Physical Review Letters, 115(19), 198501.
>
> **A5:**
> The MPTrj dataset primarily contains near-equilibrium structure relaxations and lacks short-range configurations. Alexandria includes 1D and 2D scans at more extreme conditions. As noted in the introduction, MLIPs trained on near-equilibrium data are not expected to perform well in off-equilibrium regimes.
>
> Recently, datasets designed for high-pressure DFT simulations have emerged [1]. To mitigate short-range softening issues like those in [2], newer models such as MACE-MPA incorporate an explicit ZBL core repulsion potential.
>
> [1] Chen, C.-C., et al. *npj Comput. Mater.*, **11**, 191 (2025).
>
> [2] Deng, B., et al. *arXiv:2405.07105* (2024).
>
> **A6:**
> We will carefully update punctuation and writing style during revision. Thank you for the reminder.

---

### Official Review · Reviewer_2N5w · 2025-06-29

**Rating:** 5
**Confidence:** 4

**Summary:**

This paper introduces a novel open-source evaluation platform called MLIP Arena, designed to provide a fairer and more comprehensive assessment of machine learning interatomic potentials (MLIPs) in materials simulation.

**Additional Feedback:**

- The current platform evaluates the point estimate performance of model outputs, but does not yet account for model uncertainty (e.g., confidence intervals, epistemic vs. aleatoric uncertainty).

- While the paper mentions certain model failure modes (such as short-range holes and unphysical energy jumps), it lacks a systematic classification. It is recommended to establish a structured Failure Mode Library, including types of issues, triggering conditions, and reference examples. Additionally, implementing automated Failure Diagnostics reports could help identify potential failure mechanisms of a model in specific tasks.

**Dataset Code Accessibility:**

Yes

**Dataset Code Comments:**

Yes, the authors have provided annotated datasets and code, demonstrating good openness and reproducibility.

**Ethical Comments:**

No, this study does not raise major ethical concerns.

The research focuses on developing an open and transparent evaluation platform for scientifically analyzing the performance of machine learning interatomic potential (MLIP) models. It does not involve human participants, sensitive data, privacy issues, bioethical risks, or security-sensitive technologies, nor does it promote potential social discrimination, environmental harm, or misleading applications. The model training and testing data are sourced from publicly available materials science databases, and both the motivation and methodology of the study are clearly scientific and low-risk in nature.

**Ethical Considerations:**

No, there are no or only very minor ethics concerns

**Final Justification:**

I recommend an Accept (5) as the authors provided a satisfactory rebuttal that adequately addressed the initial concerns.

**Limitations Weaknesses:**

1. Limited scope in physical consistency evaluation (Section 2.1: Asymptotic Analyses). Although the authors emphasize energy smoothness (tortuosity), conservation, and symmetry, these metrics are mostly confined to simplified 1D paths (e.g., EOS or diatomic PEC). For example:
Table 1 & Figure 2: The Equation of State (EOS) curves are sampled only under iso-volume deformations, without involving structural relaxation paths or multi-body torsions.
Figure S6: The rotational symmetry test in "Rotational force equivariance" is based on fixed angles and axes with averaged MAE, which cannot fully capture gradient consistency under multidimensional transformations.
Possible improvements include: introducing local perturbations or high-dimensional structural transformations to test symmetry and conservation; using full Hessian matrix positive-definiteness to assess the physical stability of potential energy surfaces (PES); and evaluating PES consistency and continuity in amorphous or interfacial structures.

2. Insufficient coverage of dynamical behavior (Section 2.2: Stability and Reactivity). While NEB paths are used to assess vacancy migration, they only cover ideal paths in FCC/HCP crystals. The study lacks coverage of more complex or critical migration phenomena such as interface diffusion, asymmetric paths, and surface activity.
Figure 5: The “symmetry deviation” and “MAPE@max” metrics in vacancy migration NEB profiles fail to reflect the overall energy consistency along the NEB path (e.g., path stability, saddle point diversity).
Suggested enhancements include incorporating diffusion paths in polymorphic crystals, 2D materials, and heterointerfaces; replacing single-point MAPE with integrated NEB curve metrics (e.g., total area under energy curve); and introducing bifurcating paths and asymmetric excited states in NEB analysis.

3. Entropy-based out-of-distribution (OOD) generalization metrics are insightful but lack precision (Section 2.3: Distribution Shift). Differential entropy is used to quantify “surprise,” but it heavily depends on the accuracy of the training distribution approximation. The evaluation only uses energy deviation as a proxy for generalization, ignoring the acceptability of structural and dynamical behavior.
The study also lacks a systematic analysis of the relationship between entropy and structural complexity (e.g., crystal topology, defect density).
Suggestions include incorporating structural complexity metrics (e.g., Voronoi entropy, SOAP variance); jointly modeling entropy with reactivity loss, rotational MAE, and dynamical stability; and introducing probabilistic modeling of physical inconsistency (e.g., using BMA or dropout-based uncertainty estimation).

4. Limited coverage of real scientific (task-oriented) scenarios (Section 2.4: Thermodynamic Properties and Studies). Although a few representative tasks are included, the number and variety are insufficient to cover hot domains like electrochemistry, batteries, or catalysis.
Moreover, all results lack comparison with experimental data (e.g., adsorption energy vs. measured values); and the evaluation does not consider chained task effects, such as whether prediction errors propagate and impact downstream simulations.
Recommended improvements include adding benchmark cases such as OER/CO₂RR/cathode interface reconstruction in catalysis or energy storage; validating against experimental data (e.g., using TPD experiments or XPS for adsorption energies); and designing end-to-end workflow evaluations, such as: material screening → optimization → reaction simulation → stability assessment.

**Strengths Contributions:**

Traditional MLIP evaluation relies heavily on error metrics (such as MAE and RMSE) against DFT reference values, which do not effectively capture performance under extreme conditions or in real-world scientific tasks. This paper proposes a more physically consistent and task-oriented approach to assess the generalization and practical utility of MLIPs, focusing on the following four categories of metrics:
- Asymptotic behavior far from equilibrium, such as the smoothness of the potential energy surface (PES), short-range repulsion, and the conservative nature of the force field;
- Stability and reactivity under extreme conditions, using molecular dynamics (MD) simulations to evaluate behavior under high temperature and pressure (e.g., hydrogen combustion);
- Generalization under out-of-distribution (OOD) conditions, assessing energy conservation and rotational symmetry on “unseen” or anomalous structures;
- Prediction of thermodynamic properties and phenomena, such as vacancy migration, phase transitions, and gas adsorption.

---

> ### Author Rebuttal · Authors · 2025-07-31
>
> **W1:**
>
> (1D path) Thank you for the comments on the limitations of the current Arena on 1D paths. Potential energy surfaces are inherently multi-dimensional, and we agree and recognize the limitation of such simplification. Future exploration on higher dimensional PES will be important (e.g. Ramanchandran for alanine dipeptide and beyond). The systematic approach to cover diverse chemistry warrants the careful design of such benchmarks, and we anticipate the community to explore in this direction.
>
> (EOS) Birch–Murnaghan EOS does not allow simulation cells to change shape (assumption of isotropic strain), but each shrunk/expanded structure still undergoes volume-constrained ionic relaxation. Here structural transformation is not desirable (the EOS will not conform to smooth spline curves under transformation) and is avoided by keeping small volumetric strain 20% (linear strain of 6.27%, smaller than 10% adopted by large scale screening [1] and thus less likely to have structural transformation). It is challenging to know a priori in which crystals and where the structural transformation should take place without comprehensive screening on multiple phases by DFT. We appreciate the feedback and will consider structural transformation benchmarks for the future work.
>
> (Hessian) We appreciate the suggestion. The positive definite Hessian holds true for the mechanically stable crystals and molecules in the stable conformations but may not still hold for the molecules in transition states or the amorphous structures like we introduced in this work (RM24). Quantification for these cases (high-dimensional saddle points) would be challenging and we are leaving this for future experiments.
>
>
> [1] Latimer, K., Dwaraknath, S., Mathew, K., Winston, D., & Persson, K. A. (2018). Evaluation of thermodynamic equations of state across chemistry and structure in the materials project. npj Computational Materials, 4(1), 40.
>
>
> **W2:**
>
> We appreciate the reviewer’s comments. To propose metrics agnostic to DFT references, we propose two asymmetry metrics: barrier asymmetry and path asymmetry. Among these, path asymmetry integrates along the entire NEB paths and should be considered more representative of the entire FCC/HCP vacancy migration paths.
>
> In the current version of MLIP Arena, we restrict the discussion to simple NEB pathways in ideal crystals, as this allows NEB to have robust results to compare with bias universally true (for example, migration path along FCC/HCP basal plane should be strictly symmetrical). For more complicated cases, the transition state localization and minimal energy path search with naive NEB could be difficult and are the current research in the field [1, 2].
>
> For MAPE@max, the metrics falls back to DFT reference metric and going forward would be obsolete with higher quality, larger scale of DFT calculations available, e.g. CatTSnami [3]. The shapes of NEB paths are however still functional dependent and the robust way for cross-comparison will be important for future work.
>
> [1] Ramakrishnan, K., Schaaf, L. L., Lin, C., Wang, G., & Torr, P. (2025). Implicit Neural Representations for Chemical Reaction Paths. arXiv preprint arXiv:2502.15843.
>
> [2] Wang, X., Mao, Y., & Wang, Z. (2025). Machine learning approaches for transition state prediction. Chem Catalysis.
>
> [3] Wander, B., Shuaibi, M., Kitchin, J. R., Ulissi, Z. W., & Zitnick, C. L. (2024). CatTSunami: Accelerating Transition State Energy Calculations with Pre‐trained Graph Neural Networks, arXiv.
>
> **W3:**
>
> We acknowledge that energy deviation is not the only measure of generalization ability, but high energy deviation usually indicates inaccurate structural and unstable dynamical behavior in MD simulations.
>
> To clarify, differential entropy values of the simulated subset and the structures along each trajectory are computed with respect to the entire remaining MPTrj dataset, excluding the subset used for simulations. We explicitly designed this test to evaluate generalization, which requires defining a distribution over which to make comparisons. We use MPTrj as a reference distribution of materials because it is a standard dataset in the field and is part of many MLIP training datasets. Our proposed evaluation using differential entropy can be done with distributions other than training set and is not limited to only measuring data distributions. For example, one could also measure the differential entropy of structures over the course of a reaction relative to samples of the reactants. We thank the reviewer for the feedback and will consider incorporating structural complexity metrics in tandem with differential entropy.
>
>
> **W4:**
>
> Thank you for the suggestion on including benchmarks pertinent to the proposed hot domains. We will work on the continual integration of more comprehensive benchmarks to cater the community needs in the future generation of MLIP Arena. The short-term plan is to include more molecules and reaction types complementary to the current major focus on crystals, amorphous structures, 2D materials, and MOFs.
>
> **A1+A2:**
>
> We appreciate the suggestion. During the revision, we will include more statistics where appropriate and compile a failure mode library.

---

### Author Response · Authors · 2025-08-07

We extend our sincere gratitude to the reviewers for their valuable time and constructive feedback. We are grateful to see that the significance of MLIP Arena as a novel, open-source evaluation platform for machine learning interatomic potentials has been recognized. Reviewers highlighted its strengths, including the emphasis on generalization assessment, chemical reactivity, stability under extreme conditions, and practical utility beyond traditional error metrics. We also appreciate the acknowledgment of our efforts in preparing the codebase, comparing major foundational MLIPs, and offering diverse benchmark tasks. The study was deemed technically sound and well-executed.

We have carefully considered all comments and will revise the manuscript accordingly to address the identified limitations and further strengthen the paper. Following reviewer Y322’s suggestion, we have provided updated, more detailed instructions for each benchmark on our GitHub repository. We welcome any further questions and are happy to provide clarifications.

---

### Decision · Program_Chairs · 2025-09-18

**Decision:**

Accept (spotlight)

**Comment:**

This is a novel benchmark for Machine learning interatomic potentials (MLIPs), an important topic in AI for science. Different from existing benchmarks, which are tied to specific density functional theory references, the authors propose four benchmark categories that provide actionable insights agnostic to DFT reference.

All four reviewers provided comprehensive reviews.
The main criticism of several reviewers focuses on the fact that some tests consider specific chemistries and thus test only a very limited portion of the existing space of materials. However, this is extendible, and a first paper on the topic has to be focused after all.
Overall, the reviewers recognize the relevance of and innovation in this paper, and recommend (strong) acceptance.